# Reasoning Is Not Free: Robust Adaptive Cost-Efficient Routing for LLM-as-a-Judge

Wenbo Zhang[* 1]  Lijinghua Zhang[* 1]  Liner Xiang[* 1]  Hengrui Cai[1]

## Abstract

Reasoning-capable large language models (LLMs) have recently been adopted as automated judges, but their benefits and costs in LLM-as-a-Judge settings remain unclear. Through controlled comparisons between reasoning and non-reasoning judges, we show that explicit reasoning substantially improves judgment accuracy on tasks requiring structured verification (e.g., math and coding), while offering limited or even negative gains on simpler evaluations and incurring significantly *higher computational cost*. These findings motivate that reasoning should be used selectively rather than universally, with awareness of possible *distribution shift*. We propose a Robust Adaptive Cost-Efficient Routing (RACER), which dynamically selects between reasoning and non-reasoning judges under a fixed budget by formulating routing as a constrained distributionally robust optimization problem. RACER explicitly accounts for distribution shift via a KL-divergence uncertainty set, admits an efficient primal–dual algorithm, and enjoys theoretical guarantees including uniqueness of the optimal policy and linear convergence. Extensive experiments show that RACER achieves superior accuracy–cost trade-offs under distribution shift.

## 1. Introduction

As large language models (LLMs) continue to advance, reliably evaluating the quality of their outputs has become increasingly important but challenging. Owing to the high cost and limited scalability of human evaluation, the LLM-as-a-Judge paradigm has emerged as a promising alternative (Zheng et al., 2023; Li et al., 2023; Liu et al., 2023; Kim

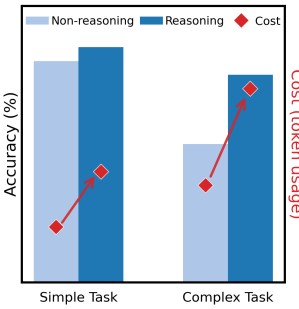
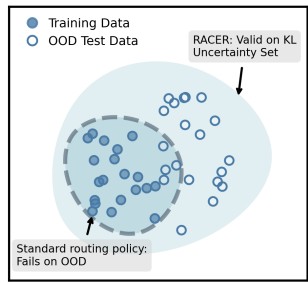

**(a)** Accuracy-cost trade-off   **(b)** Robust routing under OOD

**Figure 1.** (a) Reasoning models outperform non-reasoning models on difficult tasks, achieving higher accuracy at the cost of increased computation, while offering only marginal gains on simple tasks. (b) RACER remains robust to out-of-distribution (OOD) inputs by operating over a KL-divergence uncertainty set, whereas standard routing policies fail under distribution shift.

et al., 2023; Fu et al., 2024), in which LLMs themselves are used as evaluators. In parallel, reasoning-capable LLMs (OpenAI, 2024; Guo et al., 2025) have attracted significant attention. A growing body of work demonstrates that explicitly training models to generate long-form reasoning chains prior to producing final answers substantially improves performance across a wide range of problem-solving tasks, including mathematics, code generation, instruction following, and agentic decision-making (Yang et al., 2025; Liu et al., 2025b; Yu et al., 2025; Team et al., 2025). While these findings provide valuable insights into the role of reasoning in LLMs as problem solvers, they do not necessarily imply that such models are effective judges. In current frontier open-source models, reasoning capabilities are typically acquired via training in verifiable problem-solving tasks, whereas the judging ability is not explicitly optimized (Yang et al., 2025; Team et al., 2025; Olmo et al., 2025; Bercovich et al., 2025). This raises a crucial question:

*Can the reasoning skills learned from problem-solving tasks be effectively transferred to judgment tasks?*

Even if such transfer is possible, deploying reasoning-based judgment models introduces additional challenges. As shown in Fig. 1a, reasoning-based judgments incur substantially *higher computational cost* due to multi-step deliberation and are *not universally beneficial*, as they may

[*]Equal contribution [1]Department of Statistics, University of California, Irvine, USA. Correspondence to: Hengrui Cai <hengrc1@uci.edu>.

*Proceedings of the 43rd International Conference on Machine Learning*, Seoul, South Korea. PMLR 306, 2026. Copyright 2026 by the author(s).

induce overthinking on simple queries and lead to incorrect decisions (Chen et al., 2024c; Sui et al., 2025). A natural mitigation strategy is to learn a routing model that selectively invokes reasoning or non-reasoning models based on criteria such as expected performance or inference cost (Chen et al., 2024b; Frick et al., 2025; Liang et al., 2025; Zhang et al., 2025a). However, existing routing approaches largely overlook the issues of *distribution shift*. Routers are typically trained on static datasets, whereas real-world queries evolve due to user heterogeneity (Zhang et al., 2025c; Chakraborty et al., 2024; Son et al., 2025). Consequently, a router trained under a fixed data distribution may degrade significantly at deployment time, resulting in cost constraint violations or erroneous selections of the judging function.

To address these challenges, we propose the **Robust Adaptive Cost-Efficient Routing (RACER)**, which formulates router learning as a distributionally robust, constrained policy optimization problem, illustrated in Fig. 1b. The distributionally robust learning framework has recently been used to address the issue of distribution shift in various settings (Namkoong & Duchi, 2016; Rahimian & Mehrotra, 2019; Duchi et al., 2021). This formulation explicitly accounts for distribution shift by optimizing over an uncertainty set of data distributions centered around training data. By solving the resulting robust objective, RACER seeks policies that remain reliable under distribution shift, rather than optimizing solely for average-case performance.

Our **contributions** are summarized as follows:

• We present a comprehensive study comparing reasoning and non-reasoning modes of LLMs in LLM-as-a-Judge settings, characterizing when explicit reasoning improves judgment quality and when it fails, and providing practical insights into the effective use of reasoning for evaluation.
• We formulate router learning in a unified mathematical and algorithmic framework that jointly addresses the cost–performance trade-off and robustness to distribution shift via constrained distributionally robust optimization, followed by a tractable algorithm for efficient solution.
• We are the first in LLM routing to prove that the optimal router policy is unique and our policy iterates converge to it at a linear rate, i.e., the error contracts by a constant factor at each iteration (Wei et al., 2021; Ding et al., 2023). This provides strong theoretical support for the proposed method.
• Extensive experiments demonstrate that RACER consistently outperforms baseline methods under distribution shift, achieving improved accuracy–cost trade-offs.

## 2. Does Reasoning Help LLM-as-a-Judge?

### 2.1. Accuracy–Cost Trade-offs of Reasoning Judges

We study whether, and under what conditions, explicit reasoning improves LLM-as-a-Judge. To this end, we perform controlled comparisons between reasoning and non-reasoning judges and characterize the resulting accuracy–cost trade-offs as well as agreement patterns across benchmarks and model sizes.

**Experimental Setup.** To isolate the effect of explicit reasoning, we consider paired reasoning and non-reasoning judges $\{\mathcal{M}_{\mathrm{R}}, \mathcal{M}_{\mathrm{no\text{-}R}}\}$ and perform controlled comparisons within each pair. By matching the size, architecture, and overall capability of the model, performance differences can be primarily attributed to the presence or absence of reasoning. Concretely, for hybrid models that support both reasoning and non-reasoning inference, we instantiate the pair by treating the reasoning mode as $\mathcal{M}_{\mathrm{R}}$ and the non-reasoning mode as $\mathcal{M}_{\mathrm{no\text{-}R}}$. We evaluate multiple open-source hybrid models from the Qwen3 family (1.7B/4B/8B), as well as reasoning and non-reasoning judges on three widely used LLM-as-a-Judge benchmarks: JUDGEBENCH (Tan et al., 2025), REWARDBENCH (Lambert et al., 2025), and REWARDBENCH 2 (Malik et al., 2025), all of which consist of tasks from various domains. Each example provides a prompt, two candidate responses, and a preference label.

**Evaluation Metrics.** For evaluation, we prompt the LLM judge to compare the two responses and to provide its preferred choice. We compute *judge accuracy* as the fraction of examples for which the judge's preference matches the ground-truth label, and we record *cost* as token consumption during inference. We define ∆Accuracy as the difference in judgment accuracy between reasoning and non-reasoning modes, ∆Accuracy= $\mathrm{Acc}(\mathcal{M}_{\mathrm{R}}) - \mathrm{Acc}(\mathcal{M}_{\mathrm{no\text{-}R}})$, and define the cost ratio as the ratio of token consumption under reasoning versus non-reasoning inference.

**Results.** Fig. 2 (upper) plots ∆Accuracy against the cost ratio, where points closer to the upper-left indicate larger gains at lower additional cost. Across all three benchmarks, the strongest and most cost-effective gains concentrate in math and coding, whereas safety and knowledge show limited improvements and can even degrade under reasoning. Fig. 2 (lower) summarizes agreement patterns between reasoning and non-reasoning judges. When both modes reach the same outcome, non-reasoning is preferable due to its lower cost; when they disagree, selecting the better judge provides additional headroom. This headroom shrinks with model size, as larger non-reasoning judges already achieve stronger baseline judgment accuracy.

Overall, reasoning judge yields significant performance gains, especially in reasoning-intensive domains, illustrating that skills learned from problem-solving tasks effectively transfer to judgment tasks. Also, we notice that the reasoning should be used selectively, given its substantial cost.

Moreover, the heterogeneous accuracy–cost trade-offs in Fig. 2 suggest that both reward and compute cost can shift

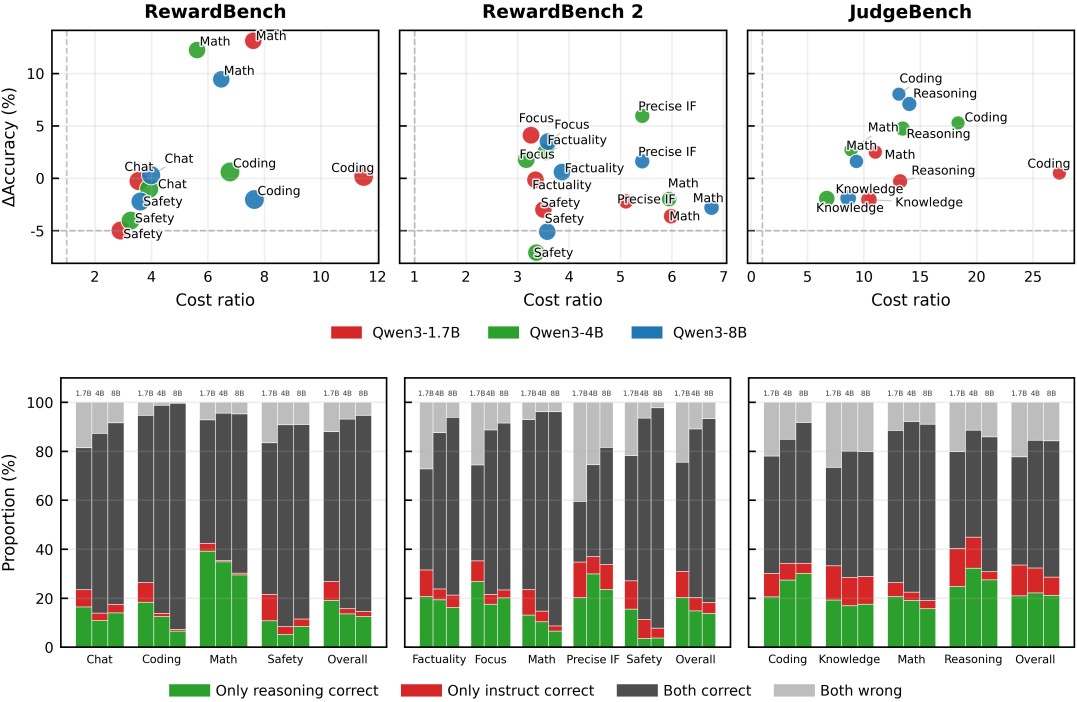

**Figure 2.** Accuracy–cost trade-offs and reasoning–instructional agreement across benchmarks. **Upper**: Accuracy improvement versus cost ratio. **Lower**: Agreement patterns between instruct and reasoning inference.

across domains and benchmarks. As a result, a router tuned on in-distribution data may mis-estimate either the benefit of reasoning or the risk of budget violation under distribution shift. This motivates our distributionally robust objective in Eq. (4), which applies robustness separately to the reward and cost terms to hedge against these two failure modes.

## 2.2. When and Why Reasoning Helps

To better understand when and why reasoning improves LLM-as-a-Judge, we conduct a case analysis. This analysis reveals three recurring patterns: (i) reasoning improves judgment when evaluation requires explicit verification; (ii) reasoning is largely redundant when the correct choice is apparent from surface cues; and (iii) reasoning can hurt when it over-expands the evaluation scope and introduces irrelevant considerations. Representative examples are summarized in Table C.1 in the Appendix.

In math and coding tasks, where correct evaluation often requires checking intermediate steps, validating internal consistency, or reconciling multiple criteria, reasoning-mode judges are better aligned with these requirements. In contrast, for tasks such as factual recall and concise question answering, reasoning and non-reasoning judges usually agree, leaving little room for improvement. Finally, we observe failure cases where explicit reasoning introduces irrelevant factors, leading to degraded judgments.

Taken together, these case-level patterns help explain the heterogeneous accuracy–cost trade-offs observed in Section 2.1. They provide qualitative evidence that reasoning improves judgment primarily when its inductive bias aligns with the evaluation structure of the task, reinforcing the conclusion that reasoning should be activated selectively rather than unconditionally under computing constraints.

## 3. RACER: Robust Adaptive Cost-Efficient Routing

### 3.1. Notation

We denote the prompt space by $\mathcal{X}$, the response space by $\mathcal{Y}$. Let $\mathcal{Z} = \mathcal{X} \times \mathcal{Y} \times \mathcal{Y}$. We use $\rho \in \Delta(\mathcal{Z} \times \{0,1\})$ to denote the distributions over pairwise prompt–response tuples with preference labels where $\Delta(\cdot)$ is a probability simplex. For a given pairwise prompt–response tuple $(x, y_1, y_2)$, the label $l = 1$ indicates that $y_1$ is preferred to $y_2$, denoted as $y_1 \succ y_2$, while $l = 0$ indicates that $y_2 \succ y_1$. We define the judge function of the model $a$ as $\Phi_a : \mathcal{Z} \mapsto \{0, 1\}$, which maps $z = (x, y_1, y_2)$ to a binary preference label, indicating whether $y_1 \succ y_2$ ($\Phi_a(z) = 1$) or $y_2 \succ y_1$ ($\Phi_a(z) = 0$). Here we assume deterministic judgments for simple cases, but the framework readily extends to stochastic settings. We denote the empirical distribution of the dataset with size $n$ as $\rho_n = \frac{1}{n} \sum_{i=1}^{n} \delta_{(z_i, l_i)}$, where $(z_i, l_i) \sim \rho$.

## 3.2. Proposed Method: RACER

Motivated by the accuracy–cost trade-offs of reasoning judges, we propose to learn a router that selectively activates reasoning modes. We formulate router learning as a constrained optimization problem. We define router policy $\pi : \mathcal{Z} \to \Delta(0, 1)$, and denote the sampled decision from $\pi(\cdot \mid z)$ as $a$ taking a value of $1$ if choosing the reasoning mode and $0$ if choosing the non-reasoning mode. Our optimization objective in terms of the budget $C$ is given by:

$$\max_{\pi \in \Pi} \mathbb{E}_{(z,l) \sim \rho_n, a \sim \pi(\cdot|z)}[r(z, a, l)],$$
$$\text{s.t.} \ \mathbb{E}_{(z,l) \sim \rho_n, a \sim \pi(\cdot|z)}[c(z, a)] \leq C, \quad (1)$$

where $r(z, a, l) = \mathbb{I}(\Phi_a(z) = l)$ with indicator function $\mathbb{I}(\cdot)$ and $c(z, a)$ denotes *whether the judgment is correct* and the *cost* of using the selected mode, respectively, and $\Pi$ is the policy class of interest. We call this method Adaptive Cost-Efficient Routing (ACER). The core idea here is that the routing policy should maximize judgment reward while controlling cost within a budget, yielding an adaptive accuracy–cost trade-off.

However, ACER is not explicitly designed to handle distribution shifts, which are common in real-world scenarios. To address this limitation, we further propose **Robust Adaptive Cost-Efficient Routing (RACER)**, which not only optimizes performance under cost constraints, but also remains robust to distribution shifts through distributionally robust learning.

We reformulate the problem as follows:

$$\max_{\pi \in \Pi} \min_{\tilde{\rho} \in \mathcal{U}(\rho_n, \delta)} \mathbb{E}_{(z,l) \sim \tilde{\rho}, a \sim \pi(\cdot|z)}[r(z, a, l)],$$
$$\text{s.t.} \ \max_{\tilde{\rho} \in \mathcal{U}(\rho_n, \delta)} \mathbb{E}_{(z,l) \sim \tilde{\rho}, a \sim \pi(\cdot|z)}[c(z, a)] \leq C. \quad (2)$$

This optimization is over a distributional uncertainty set within a distance of at most $\delta$ with $\rho_n$. Here, we utilize the uncertainty set $\mathcal{U}(\rho_n, \delta) = \{\tilde{\rho} : D_{\mathrm{KL}}(\tilde{\rho} \| \rho_n) \leq \delta\}$ and $D_{\mathrm{KL}}$ to denote the KL-divergence. Overall, RACER seeks to maximize the worst-case reward under a worst-case cost constraint, explicitly incorporating distributional robustness to prompt and response shifts, an aspect largely overlooked by existing routing methods. We now turn to solving (2).

We define the worst-case reward $R_{\mathcal{U}(\rho_n, \delta)}(\pi)$ and the worst-case cost $C_{\mathcal{U}(\rho_n, \delta)}(\pi)$ as

$$R_{\mathcal{U}(\rho_n, \delta)}(\pi) := \min_{\tilde{\rho} \in \mathcal{U}(\rho_n, \delta)} \mathbb{E}_{(z,l) \sim \tilde{\rho}, a \sim \pi(\cdot|z)}[r(z, a, l)],$$
$$C_{\mathcal{U}(\rho_n, \delta)}(\pi) := \max_{\tilde{\rho} \in \mathcal{U}(\rho_n, \delta)} \mathbb{E}_{(z,l) \sim \tilde{\rho}, a \sim \pi(\cdot|z)}[c(z, a)].$$

Hence, the original problem (2) can be rewritten as:

$$\max_{\pi \in \Pi} R_{\mathcal{U}(\rho_n, \delta)}(\pi) \quad \text{s.t.} \ C_{\mathcal{U}(\rho_n, \delta)}(\pi) \leq C. \quad (3)$$

Then the Lagrangian min-max problem associated with (3) is given by:

$$\max_{\pi \in \Pi} \min_{\lambda \geq 0} L(\pi, \lambda) := \max_{\pi \in \Pi} \min_{\lambda \geq 0} R_{\mathcal{U}(\rho_n, \delta)}(\pi) - \lambda C_{\mathcal{U}(\rho_n, \delta)}(\pi),$$

where $\lambda$ is the Lagrange multiplier. While this formulation is solvable through alternating gradient descent methods, it is computationally challenging since we do not have direct control over the data distribution $\tilde{\rho} \in \mathcal{U}(\rho_n, \delta)$ as they are not parameterized distributions. Moreover, the training data are sampled only from the source distribution $\rho$, without samples from other distributions in the uncertainty set $\mathcal{U}(\rho, \delta)$. To overcome this challenge, we introduce principled tractable algorithms to solve this problem.

The following theorem shows that the worst-case probability distribution within a KL uncertainty set can be efficiently approximated. Similar ideas have appeared in prior works on distributionally robust reinforcement learning (Gadot et al., 2024; Xu et al., 2026).

**Theorem 3.1.** *Let $\rho_n \in \Delta_n$ be the empirical distribution over $\{z_i, l_i\}_{i=1}^n$, and define $f_i := \mathbb{E}_{a \sim \pi(\cdot|z_i)}[f(z_i, a)]$. Let*

$$\mathcal{U}(\rho_n, \delta) := \Big\{ \tilde{\rho} \in \Delta_n : \sum_{i=1}^n \tilde{\rho}(i) \log \frac{\tilde{\rho}(i)}{\rho_n(i)} \leq \delta \Big\}.$$

*Define $\underline{\rho}$ and $\overline{\rho}$ as solutions to*

$$\underline{\rho} \in \arg \min_{\tilde{\rho} \in \mathcal{U}(\rho_n, \delta)} \sum_{i=1}^n \tilde{\rho}(i) f_i, \ \overline{\rho} \in \arg \max_{\tilde{\rho} \in \mathcal{U}(\rho_n, \delta)} \sum_{i=1}^n \tilde{\rho}(i) f_i.$$

*Then there exist $\underline{s}, \overline{s} \in \mathbb{R}$ and $\tau > 0$ such that, for all $i$,*

$$\underline{\rho}(i) \propto \rho_n(i) \exp\left(\frac{\underline{s} - f_i}{\tau}\right), \ \overline{\rho}(i) \propto \rho_n(i) \exp\left(\frac{f_i - \overline{s}}{\tau}\right),$$

*with $\underline{s} \leq \sum_{i=1}^n \rho_n(i) \bar{f}_i$ and $\overline{s} \geq \sum_{i=1}^n \rho_n(i) \bar{f}_i$.*

We leave the proof of Theorem 3.1 to the Appendix A.1. Based on this Theorem, we get a closed-form solution of $\underline{\rho}, \overline{\rho}$ over KL uncertainty set $\mathcal{U}(\rho_n, \delta)$. The resulting distributions $\underline{\rho}$ and $\overline{\rho}$ are obtained by reweighting the data. In the minimization setting, where $f$ is a reward function, $\underline{\rho}$ downweights samples whose expected reward exceeds the baseline $\underline{s}$ and upweights those with lower-than-baseline rewards. In the maximization setting, where $f$ is a cost function, $\overline{\rho}$ upweights samples with higher-than-baseline cost, focusing optimization on high-risk regions. The parameter $\tau$ controls the strength of reweighting: lower $\tau$ induces a more aggressive concentration on extreme (worst-case) samples.

Since the baselines $\underline{s}$ and $\overline{s}$ are generally unknown, in practice we approximate them using the empirical mean, which serves as a valid surrogate by providing upper and lower

**Algorithm 1** RACER

**input:** Number of iterations $T$, temperature $\tau$, regularization parameter $\beta$, distribution $\rho$ for preference data generation, judging functions $(\Phi_0, \Phi_1)$ by an LLM.

1: Initialize $\pi_0$ and $\lambda_0$.
2: **for** Iteration $t = 0, 1, \ldots, T-1$ **do**
3:    **Construct** $\mathcal{D}_t = \{(z, l)\}$ where $(z, l) \sim \rho$.
4:    **Sample routing actions:** $a \sim \pi_t(\cdot \mid z), \forall z \in \mathcal{D}_t$ (*optional: enumerate all actions*).
5:    **Calculate reward and cost:** For each $(z, l) \in \mathcal{D}_t$ and routing choice $a$, get judge results $\Phi_a(z)$, and calculate the reward $r(z, a, l)$ and cost $c(z, a)$.
6:    **Data reweighting:** Calculate batch mean reward $\overline{r} = \frac{\sum_{j=1}^{|\mathcal{D}_t|} r(z_j, a_j, l_j)}{|\mathcal{D}_t|}$ and cost $\overline{c} = \frac{\sum_{j=1}^{|\mathcal{D}_t|} c(z_j, a_j)}{|\mathcal{D}_t|}$, then approximate worse-case distributions:
$$\underline{\rho}(i) \propto \exp\left(\frac{\overline{r} - r_i}{\tau}\right), \bar{\rho}(i) \propto \exp\left(\frac{c_i - \overline{c}}{\tau}\right).$$
7:    **Primal-dual update:** Obtain $\pi_{t+1}$ and $\lambda_{t+1}$ by (6) and (7).
8: **end for**
9: **return** The best valid $\pi_{0:T}$ on the validation data.

---

bounds for $\underline{s}$ and $\overline{s}$, respectively. Thus, the Lagrangian min-max problem of (3) can be transformed as:

$$\max_{\pi \in \Pi} \min_{\lambda \geq 0} L(\pi, \lambda) = \max_{\pi \in \Pi} \min_{\lambda \geq 0} R_{\mathcal{U}(\rho_n, \delta)}(\pi) - \lambda C_{\mathcal{U}(\rho_n, \delta)}(\pi)$$
$$= \max_{\pi \in \Pi} \min_{\lambda \geq 0} R_{\underline{\rho}}(\pi) - \lambda C_{\overline{\rho}}(\pi). \quad (4)$$

where the last equality is implied by Theorem 3.1.

We introduce a regularized Lagrangian:

$$L_\beta(\pi, \lambda) := L(\pi, \lambda) + \beta \left( \mathcal{H}(\pi) + \frac{1}{2}\lambda^2 \right), \quad (5)$$

where $\mathcal{H}(\pi) + \frac{1}{2}\lambda^2$ is added as a regularization to the original Lagrangian $L(\pi, \lambda)$. Here $\beta$ is a regularization parameter, and $\mathcal{H}(\pi) := \mathbb{E}_{(z,l)\sim\rho}[\mathcal{H}(\pi(\cdot \mid z)]$ is the entropy of the policy $\pi$. This step allows us to control the randomness of the routing policy, thereby encouraging exploration and facilitating convergence (Cen et al., 2022; Ding et al., 2023).

We can then use the primal-dual method to solve (5):

$$\pi_{t+1} = \operatorname*{argmax}_{\pi \in \Pi} \left\{ R_{\underline{\rho}}(\pi) - \lambda_t C_{\bar{\rho}}(\pi) + \beta \mathcal{H}(\pi) \right\}, \quad (6)$$

$$\lambda_{t+1} = \operatorname*{argmax}_{\lambda \geq 0} \left\{ -\lambda_t C_{\bar{\rho}}(\pi) + \frac{1}{2}\beta\lambda_t^2 \right\}. \quad (7)$$

Based on the derivations, we propose a practical **RACER** algorithm in Algorithm 1.

## 4. Theoretical Results

In this section, we first show that the optimization problem admits a unique solution, and then demonstrate linear convergence in terms of the KL divergence between the last iterate and the optimal router policy. While recent works study routing strategies for LLMs with a focus on practical model selection and performance–cost tradeoffs (Ong et al., 2025; Liang et al., 2025; Zhang et al., 2025a), these approaches are primarily empirical and heuristic in nature. To the best of our knowledge, this work is the *first* to provide theoretical guarantees on *the convergence behavior of an LLM routing policy*.

### 4.1. Uniqueness of the Optimal Router Policy

The policy optimization problem can be interpreted as finding a saddle point of the following max–min problem,

$$\max_{\pi \in \Pi} \min_{\lambda \geq 0} \mathcal{L}_\beta(\pi, \lambda). \quad (8)$$

After establishing feasibility (i.e., the constraint set is nonempty) and showing that the solution space is bounded, we then prove that the saddle point exists and is unique, as stated in Theorem 4.1. All proofs are provided in Appendix A.2.

**Theorem 4.1** (Existence and Uniqueness of the Saddle Point). *There exists a unique pair $(\pi^*, \lambda^*) \in \Pi \times \Lambda$ such that $\mathcal{L}_\beta(\pi, \lambda^*) \leq \mathcal{L}_\beta(\pi^*, \lambda^*) \leq \mathcal{L}_\beta(\pi^*, \lambda)$ for any $\pi \in \Pi$ and $\lambda \in \Lambda$, that is, $(\pi^*, \lambda^*)$ is the saddle point of $\mathcal{L}_\beta(\pi, \lambda)$.*

As a consequence, for any $(\pi, \lambda) \in \Pi \times \Lambda$,

$$\mathcal{L}(\pi, \lambda^*) - \beta\mathcal{H}(\pi) \leq \mathcal{L}(\pi^*, \lambda^*) \leq \mathcal{L}(\pi^*, \lambda) + \frac{\beta}{2}\lambda^2,$$

which indicates that $(\pi^*, \lambda^*)$ is a saddle point of the original Lagrangian $\mathcal{L}(\pi, \lambda)$, up to two $\beta$-regularization terms.

Theorem 4.1 identifies a unique target for the primal–dual iterates. In the next subsection, we show that the updates (6) and (7) converge to this unique saddle point.

### 4.2. Convergence of the Router Policy

We next analyze the last-iterate convergence of our router policy $\pi_t$ to the optimal policy $\pi^*$. To quantify the convergence, we measure the distance between $\pi_t$ and $\pi^*$ under the distribution $\rho$ using the KL divergence, defined as

$$\mathrm{KL}(\pi_t \| \pi^*) = \mathbb{E}_{(z,l)\sim\rho} \mathrm{KL}(\pi_t(\cdot|z) \| \pi^*(\cdot|z)),$$

where

$$\mathrm{KL}(\pi_t(\cdot|z) \| \pi^*(\cdot|z)) = \sum_{a=0}^{1} \pi_t(a|z) \log\left(\frac{\pi_t(a|z)}{\pi^*(a|z)}\right).$$

To derive a closed-form solution for (6), we rewrite it as

$$\pi_{t+1} = \arg\max_{\pi \in \Pi} \mathbb{E}_{(z,l) \sim \rho, a \sim \pi(\cdot|z)} \left[ \frac{p_{\overline{\rho}}}{p_\rho} r(z, a, l) - \lambda_t \frac{p_{\overline{\rho}}}{p_\rho} c(z, a) \right]$$
$$+ \beta \, \mathbb{E}_{(z,l) \sim \rho} \left[ \mathcal{H}\left( \pi(\cdot \mid z) \right) \right], \quad (9)$$

where $p_{\overline{\rho}}$, $p_{\underline{\rho}}$, and $p_\rho$ denote the densities of the distributions $\overline{\rho}$, $\underline{\rho}$, and $\rho$, respectively.

We then propose Assumptions 1, 2, and 3 to establish the convergence result in Theorem 4.2.

**Assumption 1** (Convexity of the policy class $\Pi$). The interested policy class $\Pi$ is convex, i.e., for any $\pi_1, \pi_2 \in \Pi$ and any $\alpha \in [0, 1]$, $\alpha\pi_1 + (1 - \alpha)\pi_2 \in \Pi$.

**Assumption 2** (Boundness of $c$). There exists a constant $M > 0$, such that $c(z, a) \leq M$ for all $z \sim \rho$ and $a \in \{0, 1\}$.

**Assumption 3** (Boundness of density ratio). There exists a constant $K > 0$, such that $p_{\overline{\rho}}/p_\rho \leq K$ for all $z \in \mathcal{Z}$.

Assumption 1 is standard and has been widely adopted in convex optimization and policy optimization analyses (see, e.g., Mutti et al., 2023; Ding et al., 2023). Assumption 2 ensures that the cost $c(z, a)$ is bounded and has bounded variance, while Assumption 3 controls the amplification induced by importance reweighting. Together, these conditions imply that the dual gradient term $(p_{\overline{\rho}}/p_\rho) c(z, a)$ is uniformly bounded. This uniform bound allows us to control how much the policy $\pi_{t+1}$ responds to changes in the dual variable $\lambda_t$, which is a key ingredient in establishing the linear convergence result below.

**Theorem 4.2** (Linear Convergence of RACER). *Under Assumptions 1, 2, and 3, the router policy iterates satisfy*

$$\mathrm{KL}(\pi_t \| \pi^*) \leq \frac{M^2 K^2}{2\beta^2} \left( \frac{M^2 K^2}{M^2 K^2 + 2\beta^2} \right)^{2t} (\lambda_0 - \lambda^*)^2.$$

Theorem 4.2 establishes that the iterates $\pi_t$ converge to the optimal policy $\pi^*$ at a linear rate, depending on $M$, $K$, and $\beta$. Details and the proof are deferred to Appendix A.3. Importantly, our convergence results can be extended to the parameterized (non-convex) setting under additional assumptions on the approximation quality of the policy class $\Pi$. In particular, if each policy update approximates the optimal entropy-regularized solution up to a controlled error, then the last-iterate convergence result continues to hold up to an additional approximation error term, as commonly studied in prior work (Ding et al., 2023; Zhan et al., 2023).

# 5. Experiments

## 5.1. Experimental Setup

We conduct two complementary sets of experiments with a shared training and judgment generation protocol. The

**Table 1.** Dataset statistics and cost ratios.

| Subset | Split | # Pairs | Cost Ratio |
|---|---|---|---|
| Magpie Ultra | Train 1 | 27,785 | 11.2 |
| WildGuardMix | Train 2 | 6,709 | 3.4 |
| OffsetBias | OOD Test | 8,504 | 4.7 |

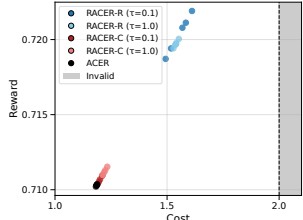 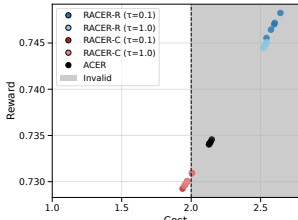

**(a)** Trained on Magpie Ultra and evaluated on OffsetBias, exhibiting a shift toward **lower-cost queries**.

**(b)** Trained on WildGuardMix and evaluated on OffsetBias, exhibiting a shift toward **higher-cost queries**.

**Figure 3.** Reward–cost trade-offs evaluated on the OOD dataset with a budget of 2. (a) shows that reward-robust RACER-R improves performance when the cost constraint is easily satisfied. (b) illustrates the importance of cost-robust RACER-C when the cost constraint can be violated under distribution shift.

first focuses on controlled ablations designed to isolate the roles of reward and cost robustness under targeted distribution shifts. The second evaluates end-to-end routing performance on real world LLM-as-a-Judge benchmarks to assess generalization under more realistic evaluation distributions.

Across both experimental settings, we use a unified data generation protocol to learn a router from the preference data. Given any preference dataset consisting of a prompt and a response pair $(x_i, y_{i,1}, y_{i,2})$, we prompt an LLM judge under two inference modes (instruct and reasoning) to select its preferred response. For each instance, we record (i) whether the judge's decision matches the ground-truth preference label, $r_i$, and (ii) the number of tokens consumed under each inference mode, $c_i$. The router takes as input a text embedding of the concatenated context $z_i = (x_i, y_{i,1}, y_{i,2})$ obtained from an embedding model.

Details of the judging prompts, decoding configurations, and the embedding model are provided in Appendix B. All training and evaluation sets in the following experiments are constructed using this protocol, differing only in the underlying dataset.

## 5.2. Ablating Reward and Cost Robustness

As shown in our robust objective (3), the proposed formulation incorporates two levels of robustness: robustness applied to the reward, denoted as $R_{\mathcal{U}(\rho_n, \delta)}(\pi)$, and robustness applied to the cost, denoted as $C_{\mathcal{U}(\rho_n, \delta)}(\pi)$. To better understand the contribution of each component, we conduct

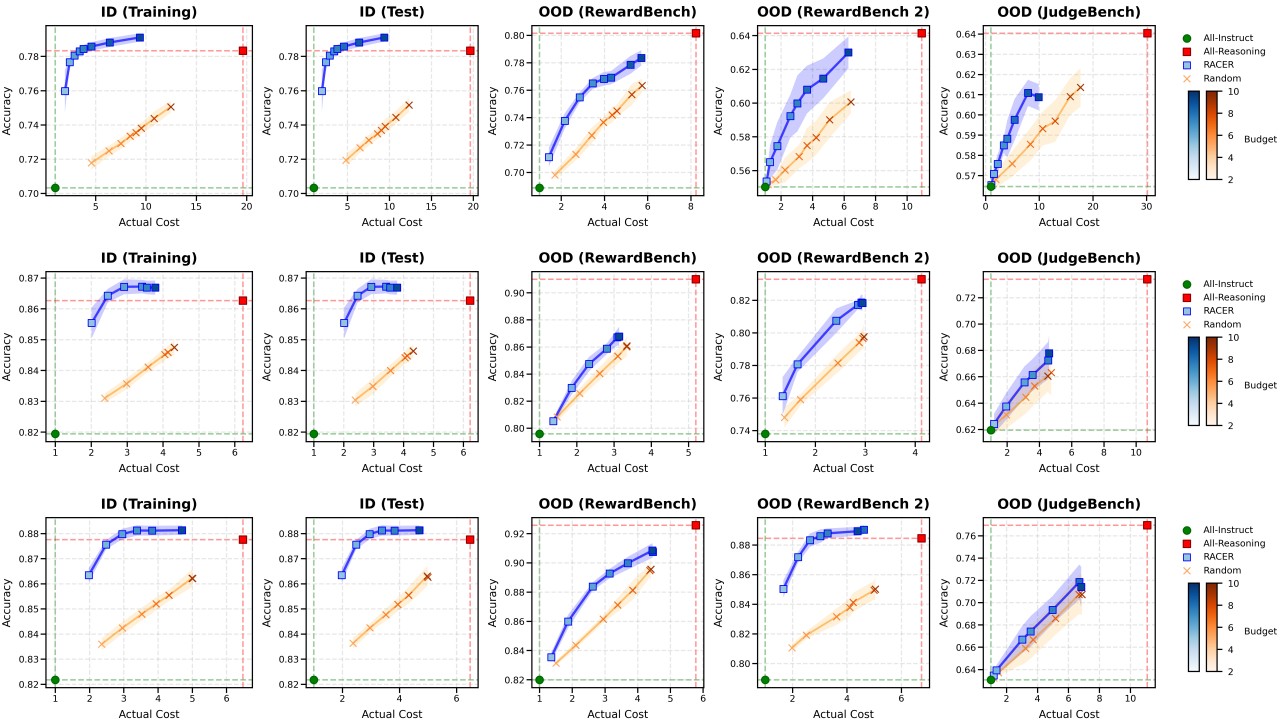

**Figure 4.** Routing performance across compute budgets on ID and OOD benchmarks (top to bottom: Qwen3-1.7B, 4B, and 8B).

an ablation study to investigate (i) when reward robustness is necessary and (ii) when cost robustness is necessary. Specifically, we introduce two variants, RACER-R and RACER-C, which apply distributionally robust reweighting only to the reward and only to the cost, respectively. We also include ACER with the non-robust objective in (1) as a baseline.

We consider two representative scenarios: (a) the OOD queries require less computation than in-distribution queries, and (b) the OOD queries require more computation than in-distribution queries. We construct our datasets from the SKYWORK REWARD PREFERENCE DATASET (Liu et al., 2024), with Magpie-Ultra and WildGuardMix used for training in Scenarios (a) and (b), respectively, and OffsetBias as the OOD test set. Judgments are produced by the hybrid reasoning model Qwen3-4B (Yang et al., 2025). Reward is defined as correctness, and cost as the relative token usage between reasoning and non-reasoning judgments (Table 1). We use bge-3 (Chen et al., 2024a) to get embeddings of the query and responses, and a linear router.

As shown in the left panel of Figure 3, the OOD shift favors lower-cost queries, all methods satisfy the budget constraint across runs. RACER-R consistently achieves the highest reward, with stronger robustness (lower temperature) yielding further gains, indicating that when OOD cost is low, emphasizing reward robustness enables more effective budget utilization. In contrast, in the right panel, RACER-C attains a lower reward than RACER-R and ACER but consistently

maintains OOD costs within budget, demonstrating that cost robustness provides a safer strategy when in-distribution cost is low but OOD shifts increase violation risk. Overall, RACER variants exhibit greater robustness in both reward and cost compared to the non-robust baseline ACER.

### 5.3. Scaling Evaluation to Standard Benchmarks

**Training data.** Following the data-generation protocol, we construct the training set from (i) 20,000 instances from SKYWORK-REWARD (Liu et al., 2024), and (ii) an additional 20,000 instances sampled from MATH-STEP-DPO-10K (Lai et al., 2024) and CODE-PREFERENCE-PAIRS (Vezora, 2024). We include the math and code data to improve coverage of reasoning-intensive domains, which are under-represented in SKYWORK-REWARD. We generate judgments by pairing the reasoning and non-reasoning variants of Qwen3-1.7B/4B/8B (Yang et al., 2025). We also include Llama-3.1-8B (Grattafiori et al., 2024) as an additional model family; the corresponding results, reported in Appendix C.3, exhibit a similar trend, indicating that RACER generalizes across model families.

**Evaluation datasets.** We evaluate the learned routing policy on standard LLM-as-a-Judge benchmarks spanning diverse task domains, including REWARDBENCH (Lambert et al., 2025), REWARDBENCH-2 (Malik et al., 2025), and JUDGEBENCH (Tan et al., 2025). All evaluation benchmarks are held out from training and are used exclusively

for out-of-distribution evaluation. Judgments are generated using the same model pairs.

**Baselines.** We compare against three baselines:
• **All-Instruct**, which always selects the instruct mode (cost fixed to 1 and accuracy equal to the instruct judge's average correctness);
• **All-Reasoning**, which always selects the reasoning mode (accuracy and cost given by the reasoning judge's average correctness and token consumption, respectively);
• **Random**, which activates the reasoning mode independently for each instance with probability equal to the learned policy's average reasoning rate.

We evaluate the routing policy under compute budgets $C \in \{2, 2.5, 3, 3.5, 4, 5, 7, 10\}$. For each budget, we repeat training and evaluation 20 times and report the mean and standard deviation of accuracy and realized cost ratio. Our routing policy is parametrized by a four-layer neural network. See Appendix B.3 and B.5 for detailed implementation and hyperparameter choices.

Fig. 4 reports accuracy–cost trade-offs under varying budgets for three model scales. On both the training split and the in-distribution test split, RACER achieves a notably favorable regime: at roughly half of the All-Reasoning cost, it matches and often surpasses the All-Reasoning judge's accuracy, indicating that reasoning can be concentrated on the subset of instances where it delivers the largest marginal benefit. Under OOD benchmarks, RACER continues to yield consistent improvements over the baselines, demonstrating that the learned selection rule generalizes beyond the training distribution. The Random baseline forms an approximately linear interpolation between the All-Instruct and All-Reasoning endpoints, corresponding to indiscriminate activation of reasoning at a fixed rate. In contrast, RACER traces a concave, higher-accuracy frontier; the area between the RACER curve and the Random curve reflects the additional gain from strategic instance-level selection, which cannot be explained by merely increasing the overall reasoning rate, but instead by allocating reasoning to the right examples under the same budget.

### 5.4. Additional Results

**Table 2.** Accuracy and cost comparison across model scales with advanced baselines.

| Method | 1.7B | | 4B | | 8B | |
|---|---|---|---|---|---|---|
| | Acc. | Cost | Acc. | Cost | Acc. | Cost |
| RouterBench-KNN | 71.3 | 2.6 | 84.1 | 2.5 | 86.8 | 2.6 |
| RouteLLM-MF | 69.4 | 3.8 | 84.7 | 3.4 | 88.2 | 4.1 |
| M-IRT | 71.6 | 3.4 | 84.3 | 2.7 | 88.9 | 3.4 |
| RACER | **72.2** | 3.6 | **85.8** | 3.4 | **90.0** | 3.9 |

**Comparison with other advanced baselines.** We also com-

**Table 3.** Sensitivity of RACER to $\beta$ under different budgets using data generated by the Qwen3-4B judge pair.

| $\beta$ | $C = 2$ | | $C = 3$ | | $C = 4$ | |
|---|---|---|---|---|---|---|
| | Acc. | Cost | Acc. | Cost | Acc. | Cost |
| 0 | 85.2 | 1.9 | 86.7 | 2.9 | 86.8 | 3.7 |
| 0.005 | 85.5 | 2.0 | 86.7 | 2.9 | 86.7 | 3.6 |
| 0.01 | 85.5 | 2.0 | 86.7 | 2.9 | 86.7 | 3.8 |
| 0.05 | 84.8 | 2.0 | 86.0 | 2.9 | 86.2 | 3.8 |

pare RACER with three representative advanced baselines: RouterBench-KNN (Hu et al., 2024), RouteLLM-MF (Ong et al., 2025), and M-IRT (Song et al., 2025), under the same budget constraint ($C = 4$). The results are averaged on 3 test datasets and over 10 replications. As shown in Table 2, RACER achieves the highest accuracy across all three model families while remaining within the budget. For the 1.7B, 4B, and 8B models, RACER improves over the strongest baseline by 0.64, 1.10, and 1.06 percentage points, respectively. These results suggest that RACER exploits the available budget more effectively by allocating reasoning to instances where it is most beneficial, yielding a stronger reward–cost trade-off than prior routing methods. Since RACER is a general framework, in principle, it can be combined with more advanced routing architectures to improve performance further.

**Ablation and Sensitivity Analysis of Entropy Regularizer.** We conduct an ablation and sensitivity analysis on $\beta$ (weight for entropy regularization) at 3 representative budget levels, using Qwen3-4B judge pair. As shown in Table 3, removing the entropy regularizer hurts accuracy under the tight budget, confirming its contribution when the constraint is binding. Performance remains stable for $\beta \in \{0.005, 0.01\}$, while $\beta = 0.05$ consistently degrades accuracy, supporting our default choice of $\beta$.

## 6. Related Work

**LLM-as-a-Judge and Reasoning.** Human evaluation is considered the gold-standard metric for assessing LLM-generated content (Ouyang et al., 2022; Zheng et al., 2023). However, it is costly and time-consuming, making it difficult to scale in practice. To address this, LLM-as-a-Judge has been proposed as an automatic proxy (Zheng et al., 2023; Li et al., 2023; Liu et al., 2023; Kim et al., 2023; Fu et al., 2024). One advantage is that the LLM can provide explanations for its final judgments, which facilitates error analysis. To evaluate how well LLM-as-a-Judge truly performs, benchmarks have been introduced to measure the accuracy of judgments across different domains (Liu et al., 2025a; Tan et al., 2025; Lambert et al., 2025). There are several recent studies have shown that incorporating reasoning into models through reinforcement learning on judge-specific tasks can further

enhance evaluation performance (Whitehouse et al., 2025; Saha et al., 2025; Chen et al., 2025; 2026). Notably, our work demonstrates that even without judge-specific training, reasoning abilities acquired from general-domain training can still effectively transfer.

**LLM Routing.** As the number of parameters in LLMs continues to grow, their inference cost increases substantially. To address this issue, LLM routing has emerged as an effective strategy that assigns user queries to appropriate LLMs while balancing both performance and cost. Chen et al. (2024b) adopted a cascading strategy, sequentially querying LLMs until a satisfactory response is produced. P2L (Frick et al., 2025) employs a prompt-to-regression approach to predict a vector of Bradley–Terry coefficients, which are then used to select the optimal model. Aggarwal et al. (2024) and Ong et al. (2025) trained a binary predictor to switch between a strong and weak model. While effective, this approach requires deploying multiple LLMs simultaneously. Recent studies (Liang et al., 2025; Zhang et al., 2025a) investigate mode switching within a single hybrid reasoning model, which is most closely related to our setting. However, we introduce a principled policy learning framework that not only maximizes performance under budget constraints, but also explicitly addresses **distribution shift**, an important yet largely overlooked challenge in existing routing systems. Furthermore, all prior studies predominantly focus on question-answering tasks, whereas we present the first comprehensive investigation of routing strategies in the LLM-as-judge domain.

**Distributionally Robust Learning.** Distributionally Robust Optimization (DRO) has been extensively studied in machine learning (Shafieezadeh Abadeh et al., 2015; Deng et al., 2020), statistics (Belloni et al., 2011; Duchi & Namkoong, 2021), and operations research (Goh & Sim, 2010; Kuhn et al., 2019). It is typically formulated as a minimax problem, in which an adversary perturbs the data-generating distribution within a prescribed uncertainty set to maximize the expected loss, while the learner optimizes model parameters to minimize this worst-case risk. Among various constructions of uncertainty sets, $f$-divergence balls are a commonly used choice (Hu & Hong, 2013; Namkoong & Duchi, 2016; Levy et al., 2020; Duchi & Namkoong, 2021), due to their strong connections with classical divergence measures and well-studied statistical properties that facilitate analysis and implementation in DRO frameworks.

## 7. Conclusion and Limitation

In this work, we study whether reasoning capabilities acquired by LLMs from problem-solving tasks transfer to judgment tasks. We find that reasoning-based judges can substantially improve accuracy, but the gains are task-dependent and often incur significantly higher computational cost, par-

ticularly under distribution shift. To address these problems, we propose RACER, which adaptively activates reasoning judges under a fixed budget. We are also the first to provide a theoretical analysis with linear convergence guarantees, supporting both efficiency and robustness. Empirically, RACER consistently outperforms baselines on multiple out-of-distribution benchmarks, achieving superior accuracy–cost trade-offs.

RACER focuses on binary routing with a KL-based uncertainty set; while effective under moderate distribution shifts, overly large uncertainty can induce conservative routing that prioritizes cost safety and underutilizes reasoning. Extending RACER to richer routing schemes and alternative uncertainty sets to better balance robustness and adaptivity is an important direction for future work.

## Acknowledgements

This work was supported in part by the National Science Foundation under Grant No. DMS-2401271. We thank the anonymous reviewers for their helpful comments.

## Impact Statement

This work examines the benefits and costs of reasoning-capable LLMs in the LLM-as-a-Judge setting and proposes a robust routing framework that selectively allocates reasoning under budget and distribution shift constraints. A key positive impact is promoting more efficient and sustainable use of computation by showing that reasoning is not universally beneficial and should be applied selectively, while also improving the robustness and reliability of automated evaluation under realistic, non-stationary data distributions. These advances can support more trustworthy benchmarking and model development practices. Potential risks include increased reliance on automated judges, which may propagate biases or systematic errors inherited from the underlying models, and the possibility that adaptive routing could be misused to unevenly apply stronger evaluation. We mitigate these concerns by explicitly analyzing failure cases, prioritizing robustness over maximal performance, and positioning the method as a complement to human evaluation rather than a replacement.

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

# Appendix

The appendix provides additional theoretical analysis, experimental details, and supplementary experimental results. In Section A, we establish the existence and uniqueness of the saddle point and prove last-iterate convergence, providing theoretical guarantees for the convergence and optimality of the proposed algorithm, RACER. In Section B, we describe in detail the data generation process, the training procedure, and the evaluation protocol. In Section C, we report supplementary results, including training curves and representative case studies that further motivate our proposed algorithm.

## A. Theoretical Analysis

### A.1. Deriving Data Distribution from KL Uncertainty Set

**Lemma A.1.** *Let $\rho_n \in \Delta_n$ be the empirical distribution over $\{z_i\}_{i=1}^n$, and define $f_i := \mathbb{E}_{a \sim \pi(\cdot|z_i)}[f(z_i, a)]$. Let*

$$\mathcal{U}(\rho_n, \delta) := \left\{ \tilde{\rho} \in \Delta_n : \sum_{i=1}^n \tilde{\rho}(i) \log \frac{\tilde{\rho}(i)}{\rho_n(i)} \leq \delta \right\}.$$

*Define $\underline{\rho}$ and $\overline{\rho}$ as the solutions to*

$$\underline{\rho} \in \arg\min_{\tilde{\rho} \in \mathcal{U}(\rho_n, \delta)} \sum_{i=1}^n \tilde{\rho}(i) f_i, \qquad \overline{\rho} \in \arg\max_{\tilde{\rho} \in \mathcal{U}(\rho_n, \delta)} \sum_{i=1}^n \tilde{\rho}(i) f_i.$$

*Then there exist $\underline{s}, \overline{s} \in \mathbb{R}$ and $\tau > 0$ such that, for all $i$,*

$$\underline{\rho}(i) \propto \rho_n(i) \exp\left( \frac{\underline{s} - f_i}{\tau} \right), \qquad \overline{\rho}(i) \propto \rho_n(i) \exp\left( \frac{f_i - \overline{s}}{\tau} \right),$$

*with $\underline{s} \leq \sum_{i=1}^n \rho_n(i)\bar{f}_i$ and $\overline{s} \geq \sum_{i=1}^n \rho_n(i)\bar{f}_i$.*

*Proof.* We prove the claim for $\underline{\rho}$; the proof for $\overline{\rho}$ is analogous.

Consider the convex optimization problem

$$
\begin{aligned}
\underset{\tilde{\rho} \in \mathbb{R}^n}{\text{minimize}} \quad & \sum_{i=1}^n \tilde{\rho}(i) f_i \\
\text{subject to} \quad & \sum_{i=1}^n \tilde{\rho}(i) \log \frac{\tilde{\rho}(i)}{\rho_n(i)} \leq \delta, \\
& \sum_{i=1}^n \tilde{\rho}(i) = 1, \\
& \tilde{\rho}(i) \geq 0, \ \forall i.
\end{aligned}
\tag{A.1}
$$

The objective is linear and the KL constraint is convex. Moreover, Slater's condition holds (e.g., $\tilde{\rho} = \rho_n$ is strictly feasible when $\delta > 0$). Hence, the Karush–Kuhn–Tucker (KKT) conditions are necessary and sufficient for optimality.

Introducing Lagrange multipliers $\lambda \geq 0$ for the KL constraint, $\mu \in \mathbb{R}$ for normalization, and $\nu_i \geq 0$ for non-negativity, the Lagrangian of (A.1) is

$$\mathcal{L}(\tilde{\rho}, \lambda, \mu, \nu) = \sum_{i=1}^n \tilde{\rho}(i) f_i + \lambda\left( \sum_{i=1}^n \tilde{\rho}(i) \log \frac{\tilde{\rho}(i)}{\rho_n(i)} - \delta \right) + \mu\left( \sum_{i=1}^n \tilde{\rho}(i) - 1 \right) - \sum_{i=1}^n \nu_i \tilde{\rho}(i). \tag{A.2}$$

The stationarity condition of the KKT implies that, for all $i$,

$$f_i + \lambda\left( \log \frac{\tilde{\rho}(i)}{\rho_n(i)} + 1 \right) + \mu - \nu_i = 0. \tag{A.3}$$

Solving (A.3) for $\tilde{\rho}(i)$ yields

$$\tilde{\rho}(i) = \rho_n(i) \exp\left(-\frac{f_i + \mu + \lambda - \nu_i}{\lambda}\right). \tag{A.4}$$

In the non-degenerate case (e.g., $\delta > 0$ and $\{f_i\}$ not all equal), the KL constraint is active at the optimum, and thus $\lambda > 0$. Since $\rho_n(i) > 0$ and the exponential is strictly positive, Eq. (A.4) implies $\tilde{\rho}(i) > 0$ for all $i$. By complementary slackness, we therefore have $\nu_i = 0$ for all $i$, and Eq. (A.4) simplifies to

$$\tilde{\rho}(i) = \rho_n(i) \exp\left(-\frac{f_i + \mu + \lambda}{\lambda}\right). \tag{A.5}$$

Let $\tau := \lambda$ and define $\underline{s} := -(\mu + \lambda)$. Then Eq. (A.5) can be written as

$$\tilde{\rho}(i) \propto \rho_n(i) \exp\left(\frac{\underline{s} - f_i}{\tau}\right), \tag{A.6}$$

which establishes the stated form of $\underline{\rho}$.

It remains to show that $\underline{s} \le \sum_i \rho_n(i) f_i$. Using the normalization condition $\sum_i \tilde{\rho}(i) = 1$ and Eq. (A.6), we obtain

$$\exp\left(\frac{\underline{s}}{\tau}\right) = \frac{1}{\sum_{i=1}^{n} \rho_n(i) \exp\left(-\frac{f_i}{\tau}\right)}. \tag{A.7}$$

Applying Jensen's inequality to the convex function $\exp(\cdot)$ yields

$$\exp\left(-\frac{\sum_{i=1}^{n} \rho_n(i) f_i}{\tau}\right) \le \sum_{i=1}^{n} \rho_n(i) \exp\left(-\frac{f_i}{\tau}\right). \tag{A.8}$$

Combining (A.7) and (A.8) gives

$$\exp\left(\frac{\underline{s}}{\tau}\right) \le \exp\left(\frac{\sum_{i=1}^{n} \rho_n(i) f_i}{\tau}\right),$$

which implies $\underline{s} \le \sum_{i=1}^{n} \rho_n(i) f_i$.

The proof for $\overline{\rho}$ follows identically by applying the same argument to $\max_{\tilde{\rho} \in \mathcal{U}(\rho_n, \delta)} \sum_i \tilde{\rho}(i) f_i$ (equivalently, minimizing $-\sum_i \tilde{\rho}(i) f_i$), yielding $\overline{\rho}(i) \propto \rho_n(i) \exp((f_i - \overline{s})/\tau)$ and $\overline{s} \ge \sum_i \rho_n(i) f_i$. □

## A.2. Existence and Uniqueness of the Saddle Point (Theorem 4.1)

We first present Proposition A.2, which states the Slater condition (feasibility) and is commonly required in the duality analysis of constrained optimization (see, e.g., Ding et al., 2023; 2025).

### A.2.1. FEASIBILITY CHECK

**Proposition A.2** (Feasibility). *There exist a constant $\xi > 0$ and a policy $\pi_0 \in \Pi$ such that $\mathbb{E}_{(z,l) \sim \underline{\rho}, a \sim \pi_0(\cdot|z)}[c(z,a)] - C \le -\xi$, where $\pi_0(0 \mid z) = 1$ and $\pi_0(1 \mid z) = 0$ for all $z \in \mathcal{Z}$.*

Proposition A.2 is straightforward to verify. Since $\mathbb{E}_{(z,l) \sim \underline{\rho}, a \sim \pi_0(\cdot|z)}[c(z,a)] = 1$, the inequality holds whenever $\xi \le C - 1$ with $C > 1$. To analyze the saddle point of $\mathcal{L}_\beta(\pi, \lambda)$, we define $\pi^*$ as the optimal solution to (8), that is,

$$\pi^* \in \arg\max_{\pi \in \Pi} \min_{\lambda \ge 0} \mathcal{L}_\beta(\pi, \lambda),$$

and let $\lambda^*$ denote the corresponding optimal dual variable,

$$\lambda^* \in \arg\min_{\lambda \ge 0} \max_{\pi \in \Pi} \mathcal{L}_\beta(\pi, \lambda),$$

with $\Lambda^*$ denoting the set of all optimal dual variables. We denote $\mathcal{L}_\beta^\lambda := \max_{\pi \in \Pi} \mathcal{L}_\beta(\pi, \lambda)$. We can then show that $\Lambda^*$ is bounded as established in Lemma A.3.

### A.2.2. BOUNDNESS OF SOLUTION SPACE

**Lemma A.3** (Boundness of $\lambda^*$). *The optimal dual variable $\lambda^*$ satisfies*

$$0 \leq \lambda^* \leq \frac{\mathcal{L}_\beta^{\lambda^*} - \mathbb{P}_\rho(\phi_0(z) = l)}{\xi},$$

*and hence $\Lambda^*$ is bounded.*

*Proof.* Let $\Lambda_a := \{\lambda \geq 0 \mid \mathcal{L}_\beta^\lambda \leq a\}$ be a sublevel set of the dual objective for $a \in \mathbb{R}$. Recall that $\mathcal{L}_\beta^\lambda = \max_{\pi \in \Pi} \mathcal{L}_\beta(\pi, \lambda)$. For any $\lambda \in \Lambda_a$, we have

$$a \geq \mathcal{L}_\beta^\lambda \geq \mathcal{L}_\beta(\pi_0, \lambda)$$

$$\geq \mathbb{E}_{(z,l)\sim\rho,a\sim\pi_0(\cdot|z)}[r(z,a,l)] - \lambda \left( \mathbb{E}_{(z,l)\sim\bar\rho,a\sim\pi_0(\cdot|z)}[c(z,a)] - C \right) + \beta \left( \mathbb{E}_{(z,l)\sim\rho} [\mathcal{H}(\pi_0(\cdot \mid z))] + \frac{1}{2}\lambda^2 \right)$$

$$\geq \mathbb{E}_{(z,l)\sim\rho,a\sim\pi_0(\cdot|z)}[r(z,a,l)] + \lambda\xi + \beta \left( \mathbb{E}_{(z,l)\sim\rho} [\mathcal{H}(\pi_0(\cdot \mid z))] + \frac{1}{2}\lambda^2 \right),$$

where the last inequality comes from Proposition A.2. Because $\mathcal{H}(\pi_0(\cdot \mid z)) = 0$ and

$$\mathbb{E}_{(z,l)\sim\rho,a\sim\pi_0(\cdot|z)}[r(z,a,l)] = \mathbb{P}_\rho(\phi_0(z) = l),$$

it follows that

$$a \geq \mathcal{L}_\beta^\lambda \geq \mathbb{P}_\rho(\phi_0(z) = l) + \lambda\xi.$$

Thus we get $\lambda \leq \frac{\mathcal{L}_\beta^\lambda - \mathbb{P}_\rho(\phi_0(z)=l)}{\xi}$. Finally, setting $a = \mathcal{L}_\beta^{\lambda^*}$ with $\Lambda_a = \Lambda^*$, we derive

$$0 \leq \lambda^* \leq \frac{\mathcal{L}_\beta^{\lambda^*} - \mathbb{P}_\rho(\phi_0(z) = l)}{\xi}.$$

$\square$

Based on Lemma A.3, we define the domain for $\lambda$ as

$$\Lambda = [0, \frac{\mathcal{L}_\beta^{\lambda^*} - \mathbb{P}_\rho(\phi_0(z) = l)}{\xi}],$$

which ensures $\Lambda^* \subseteq \Lambda$. With this setup, all the required conditions for applying Sion's minimax theorem (Sion, 1958) are satisfied, which ensures the existence and uniqueness of the regularized saddle point.

### A.2.3. PROOF OF THEOREM 4.1

*Proof.* Based on Assumption 1, $\Pi$ is a convex set. Moreover, since $\Lambda$ is convex and bounded, i.e., compact, Sion's minimax theorem (Sion, 1958) guarantees the existence of a saddle point of $\mathcal{L}_\beta(\pi, \lambda)$ in $\Pi \times \Lambda$. We next prove the uniqueness of this saddle point. Recall that

$$\mathcal{L}_\beta(\pi, \lambda) = \mathbb{E}_{(z,l)\sim\rho,a\sim\pi(\cdot|z)}[r(z,a,l)] - \lambda \left( \mathbb{E}_{(z,l)\sim\bar\rho,a\sim\pi(\cdot|z)}[c(z,a)] - C \right) + \beta \left( \mathbb{E}_{(z,l)\sim\rho} [\mathcal{H}(\pi(\cdot \mid z))] + \frac{1}{2}\lambda^2 \right).$$

Since $\frac{\partial^2 \mathcal{L}_\beta(\pi,\lambda)}{\partial \lambda^2} = \beta > 0$, $\mathcal{L}_\beta(\pi, \lambda)$ is strictly convex in $\lambda$. Then we will show $\mathcal{L}_\beta(\pi, \lambda)$ is strictly concave in $\pi$. We first examine the expectation terms:

$$\mathbb{E}_{(z,l)\sim\rho,a\sim\pi(\cdot|z)}[r(z,a,l)] = \mathbb{E}_{(z,l)\sim\rho} \left( \sum_{a=0}^{1} r(z,a,l)\pi(a|z) \right),$$

and

$$\mathbb{E}_{(z,l)\sim\overline{\rho},a\sim\pi(\cdot|z)}c(z,a) = \mathbb{E}_{(z,l)\sim\overline{\rho}}\left(\sum_{a=0}^{1}c(z,a)\pi(a|z)\right).$$

Thus, both $\mathbb{E}_{(z,l)\sim\underline{\rho},a\sim\pi(\cdot|z)}[r(z,a,l)]$ and $\mathbb{E}_{(z,l)\sim\overline{\rho},a\sim\pi(\cdot|z)}c(z,a)$ are linear in $\pi$. Because the entropy term $\mathcal{H}\left(\pi(\cdot\mid z)\right)$ is strongly concave in $\pi$, it follows that its expectation $\mathbb{E}_{(z,l)\sim\rho}\left[\mathcal{H}\left(\pi(\cdot\mid z)\right)\right]$ remains strongly concave in $\pi$. Consequently, we have $\mathcal{L}_\beta(\pi,\lambda)$ is also strongly concave in $\pi$ for any fixed $\lambda$.

Now we have shown $\mathcal{L}_\beta(\pi,\lambda)$ is strictly concave in $\pi$ and strictly convex in $\lambda$, which implies saddle point $(\pi^*,\lambda^*)$ is unique.

The existence of a saddle point as guaranteed by Lemma A.3, further implies that for any $(\pi,\lambda)\in\Pi\times\Lambda$,

$$\mathcal{L}_\beta(\pi,\lambda^*) \leq \mathcal{L}_\beta(\pi^*,\lambda^*) \leq \mathcal{L}_\beta(\pi^*,\lambda).$$

The first inequality indicates that for any $\pi\in\Pi$,

$$\mathcal{L}(\pi^*,\lambda^*) + \beta\left(\mathbb{E}_{(z,l)\sim\rho}\left[\mathcal{H}\left(\pi^*(\cdot\mid z)\right)\right] + \frac{1}{2}(\lambda^*)^2\right) \geq \mathcal{L}(\pi,\lambda^*) + \beta\left(\mathbb{E}_{(z,l)\sim\rho}\left[\mathcal{H}\left(\pi(\cdot\mid z)\right)\right] + \frac{1}{2}(\lambda^*)^2\right)$$

$$\geq \mathcal{L}(\pi,\lambda^*) + \frac{\beta}{2}(\lambda^*)^2,$$

while the second inequality implies that for any $\lambda\in\Lambda$,

$$\mathcal{L}(\pi^*,\lambda) + \beta\left(\mathbb{E}_{(z,l)\sim\rho}\left[\mathcal{H}\left(\pi^*(\cdot\mid z)\right)\right] + \frac{1}{2}\lambda^2\right) \geq \mathcal{L}(\pi^*,\lambda^*) + \beta\left(\mathbb{E}_{(z,l)\sim\rho}\left[\mathcal{H}\left(\pi^*(\cdot\mid z)\right)\right] + \frac{1}{2}(\lambda^*)^2\right)$$

$$\geq \mathcal{L}(\pi^*,\lambda^*) + \beta\,\mathbb{E}_{(z,l)\sim\rho}\left[\mathcal{H}\left(\pi^*(\cdot\mid z)\right)\right].$$

Combining the two inequalities above shows that $(\pi^*,\lambda^*)$ is a saddle point of the original Lagrangian $\mathcal{L}(\pi,\lambda)$, up to two $\beta$-regularization terms. □

### A.3. Proof of Last-iterate Convergence (Theorem 4.2)

*Proof.* In practice, we will update $(\pi_t,\lambda_t)$ as the following steps.

$$\pi_{t+1} = \arg\max_{\pi\in\Pi}\mathbb{E}_{z\sim\rho,a\sim\pi(\cdot|z)}\left[r(z,a)\right] - \lambda_t\mathbb{E}_{z\sim\overline{\rho},a\sim\pi(\cdot|z)}[g(z,a)] + \beta\,\mathbb{E}_{z\sim\rho}\left[\mathcal{H}\left(\pi(\cdot\mid z)\right)\right], \tag{A.9}$$

$$\lambda_{t+1} = \left[\lambda_t + \eta\left(\mathbb{E}_{z\sim\overline{\rho},a\sim\pi_{t+1}(\cdot|z)}[g(z,a)] - C - \beta\lambda_t\right)\right]_+, \tag{A.10}$$

where $\eta$ denotes the step size.

For notation simplicity, we denote $w_1(z) = p_{\underline{\rho}}(z)/p_\rho(z)$ and $w_2(z) = p_{\overline{\rho}}(z)/p_\rho(z)$ for any $z\in\mathcal{Z}$, which are well defined under Assumption 3.

**Step 1.** We first derive the closed-form solution to (6).

According to (6), we optimize the following objective

$$\max_{\pi\in\Pi}\mathbb{E}_{(z,l)\sim\underline{\rho},a\sim\pi(\cdot|z)}[r(z,a,l)] - \lambda_t\mathbb{E}_{(z,l)\sim\overline{\rho},a\sim\pi(\cdot|z)}[c(z,a)] + \beta\,\mathbb{E}_{(z,l)\sim\rho}\left[\mathcal{H}\left(\pi(\cdot\mid z)\right)\right]. \tag{A.11}$$

Analogously to arguments used in the proof of DPO (Rafailov et al., 2023), (A.11) can be rewritten as

$$\max_{\pi \in \Pi} \mathbb{E}_{(z,l) \sim \underline{\rho}, a \sim \pi(\cdot|z)}[r(z,a,l)] - \lambda_t \mathbb{E}_{(z,l) \sim \overline{\rho}, a \sim \pi(\cdot|z)}[c(z,a)] + \beta \mathbb{E}_{(z,l) \sim \rho}[\mathcal{H}(\pi(\cdot \mid z))] \tag{A.12}$$

$$= \max_{\pi \in \Pi} \mathbb{E}_{(z,l) \sim \rho, a \sim \pi(\cdot|z)}[w_1(z)r(z,a,l) - \lambda_t w_2(z)c(z,a)] + \beta \mathbb{E}_{(z,l) \sim \rho}[\mathcal{H}(\pi(\cdot \mid z))]$$

$$= \max_{\pi \in \Pi} \mathbb{E}_{(z,l) \sim \rho, a \sim \pi(\cdot|z)}[w_1(z)r(z,a,l) - \lambda_t w_2(z)c(z,a)] - \beta \mathbb{E}_{(z,l) \sim \rho, a \sim \pi(\cdot|z)}[\log(\pi(a|z))]$$

$$= \beta \max_{\pi \in \Pi} \mathbb{E}_{(z,l) \sim \rho} \mathbb{E}_{a \sim \pi(\cdot|z)} \left[ \frac{1}{\beta} w_1(z)r(z,a,l) - \frac{\lambda_t}{\beta} w_2(z)c(z,a) - \log(\pi(a|z)) \right]$$

$$= \beta \max_{\pi \in \Pi} \mathbb{E}_{(z,l) \sim \rho} \mathbb{E}_{a \sim \pi(\cdot|z)} \left[ \log \frac{\exp\left(\frac{1}{\beta} w_1(z)r(z,a,l)\right)}{\exp\left(\frac{\lambda_t}{\beta} w_2(z)c(z,a)\right)\pi(a|z)} \right]$$

$$= -\beta \min_{\pi \in \Pi} \mathbb{E}_{(z,l) \sim \rho} \mathbb{E}_{a \sim \pi(\cdot|z)} \left[ \log \frac{\pi(a|z)}{\exp\left(\frac{1}{\beta} w_1(z)r(z,a,l) - \frac{\lambda_t}{\beta} w_2(z)c(z,a)\right)} \right]$$

$$= -\beta \min_{\pi \in \Pi} \mathbb{E}_{(z,l) \sim \rho} \mathbb{E}_{a \sim \pi(\cdot|z)} \left[ \log \frac{\pi(a|z)}{\exp\left(\frac{1}{\beta} w_1(z)r(z,a,l) - \frac{\lambda_t}{\beta} w_2(z)c(z,a)\right)\frac{1}{Q(z)}} - \log(Q(z)) \right], \tag{A.13}$$

where we have partition function:

$$Q(z, \lambda_t) = \sum_{a=0}^{1} \exp\left( \frac{1}{\beta} w_1(z)r(z,a,l) - \frac{\lambda_t}{\beta} w_2(z)c(z,a) \right).$$

Note that the partition function depends only on $z$, $\lambda_t$ and is independent of the policy $\pi$. We now define

$$\pi_{t+1}(a|z) = \frac{1}{Q(z, \lambda_t)} \exp\left( \frac{1}{\beta} w_1(z)r(z,a,l) - \frac{\lambda_t}{\beta} w_2(z)c(z,a) \right),$$

which forms a valid probability distribution. Substituting this definition into (A.13), we can rewrite the objective as

$$-\beta \min_{\pi \in \Pi} \mathbb{E}_{(z,l) \sim \rho} [\mathrm{KL}(\pi(\cdot \mid z) \| \pi_{t+1}(\cdot \mid z)) - \log(Q(z, \lambda_t))].$$

Since $Q(z, \lambda_t)$ does not depend on $\pi$, the minimum is achieved by the policy that minimizes the KL term. By Gibbs' inequality, the KL divergence attains its minimum value of zero if and only if the two distributions are identical. Therefore, the updated policy is given by

$$\pi_{t+1}(a|z) = \frac{1}{Q(z, \lambda_t)} \exp\left( \frac{1}{\beta} w_1(z)r(z,a,l) - \frac{\lambda_t}{\beta} w_2(z)c(z,a) \right). \tag{A.14}$$

**Step 2.** We next establish the convergence of $\lambda_t$.

We view $\pi_{t+1}$ as a function of $\lambda_t$, and denote it by $\widetilde{\pi}_{\lambda_t} := \pi_{t+1}$. For notational convenience, we denote $d(\lambda) := \mathcal{L}_\beta^\lambda = \max_{\pi \in \Pi} \mathcal{L}_\beta(\pi, \lambda) = \mathcal{L}_\beta(\widetilde{\pi}_\lambda, \lambda)$. Then $d(\lambda)$ can be explicitly written as,

$$d(\lambda) = \beta \mathbb{E}_{(z,l) \sim \rho}[\log(Q(z, \lambda))] + \lambda C + \frac{\beta}{2}\lambda^2$$

$$= \beta \mathbb{E}_{(z,l) \sim \rho}\left[ \log \sum_{a=0}^{1} \exp\left( \frac{1}{\beta} w_1(z)r(z,a,l) - \frac{\lambda}{\beta} w_2(z)c(z,a) \right) \right] + \lambda C + \frac{\beta}{2}\lambda^2.$$

Consequently, the update rule in (7) can be expressed as $\lambda_{t+1} = [\lambda_t - \eta \, d'(\lambda_t)]_+$. We then compute the first and second derivatives of $d(\lambda)$. Notice that

$$d'(\lambda) = \beta \left( \mathbb{E}_{(z,l) \sim \rho}[\log(Q(z, \lambda))] \right)' + C + \beta \lambda, \tag{A.15}$$

so it suffices to compute $\left(\mathbb{E}_{(z,l)\sim\rho}\left[\log(Q(z,\lambda))\right]\right)'$. We have

$$\frac{\partial Q(z,\lambda)}{\partial\lambda} = \sum_{a=0}^{1} \exp\left(\frac{1}{\beta}w_1(z)r(z,a,l) - \frac{\lambda}{\beta}w_2(z)c(z,a)\right)\left(-\frac{1}{\beta}w_2(z)c(z,a)\right),$$

and therefore,

$$\frac{\partial\log Q(z,\lambda)}{\partial\lambda} = \frac{\sum_{a=0}^{1}\exp\left(\frac{1}{\beta}w_1(z)r(z,a,l) - \frac{\lambda}{\beta}w_2(z)c(z,a)\right)\left(-\frac{1}{\beta}w_2(z)c(z,a)\right)}{\sum_{a=0}^{1}\exp\left(\frac{1}{\beta}w_1(z)r(z,a,l) - \frac{\lambda}{\beta}w_2(z)c(z,a)\right)}$$

$$= \sum_{a=0}^{1}\widetilde{\pi}_\lambda(a|z)\left[-\frac{1}{\beta}w_2(z)c(z,a)\right] = \mathbb{E}_{a\sim\widetilde{\pi}_\lambda(\cdot|z)}\left[-\frac{1}{\beta}w_2(z)c(z,a)\right].$$

Based on Assumption 2 and 3, we have $|-\frac{\lambda}{\beta}w_2(z)c(z,a)| \leq \frac{MK}{\beta}$, thus

$$\left|\frac{\partial\log Q(z,\lambda)}{\partial\lambda}\right| \leq \frac{MK}{\beta},$$

which ensures the boundedness of the derivative. Hence, the order of differentiation and expectation can be interchanged. Thus, (A.15) comes to be

$$d'(\lambda) = -\mathbb{E}_{(z,l)\sim\rho, a\sim\widetilde{\pi}_\lambda(\cdot|z)}w_2(z)c(z,a) + C + \beta\lambda.$$

For the second derivative, we have

$$d''(\lambda) = -\left(\mathbb{E}_{(z,l)\sim\rho, a\sim\widetilde{\pi}_\lambda(\cdot|z)}w_2(z)c(z,a)\right)' + \beta, \tag{A.16}$$

where

$$\left(\mathbb{E}_{a\sim\widetilde{\pi}_\lambda(\cdot|z)}w_2(z)c(z,a)\right)' = w_2(z)\sum_{a=0}^{1}\frac{\partial\widetilde{\pi}_\lambda(a|z)}{\partial\lambda}c(z,a). \tag{A.17}$$

Let $s_\lambda(a) = \exp\left(\frac{1}{\beta}w_1(z)r(z,a,l) - \frac{\lambda}{\beta}w_2(z)c(z,a)\right)$, then $\widetilde{\pi}_\lambda(a|z) = s_\lambda(a)/((s_\lambda(0) + s_\lambda(1))$. By the chain rule, we have

$$\frac{\partial\widetilde{\pi}_\lambda(a|z)}{\partial\lambda} = \sum_{b=0}^{1}\frac{\partial\widetilde{\pi}_\lambda(a|z)}{\partial s_\lambda(b)}\frac{\partial s_\lambda(b)}{\partial\lambda}. \tag{A.18}$$

We can derive that

$$\frac{\partial s_\lambda(b)}{\partial\lambda} = s_\lambda(b)\left(-\frac{1}{\beta}w_2(z)c(z,b)\right), \tag{A.19}$$

and

$$\frac{\partial\widetilde{\pi}_\lambda(a|z)}{\partial s_\lambda(b)} = \frac{\mathbb{I}(a=b)s_\lambda(1-a) - \mathbb{I}(a\neq b)s_\lambda(a)}{((s_\lambda(0) + s_\lambda(1))^2} = \frac{\mathbb{I}(a=b)\widetilde{\pi}_\lambda(1-a|z) - \mathbb{I}(a\neq b)\widetilde{\pi}_\lambda(a|z)}{s_\lambda(0) + s_\lambda(1)}. \tag{A.20}$$

Combining (A.19) and (A.20), (A.18) comes to be

$$\frac{\partial\widetilde{\pi}_\lambda(a|z)}{\partial\lambda} = \widetilde{\pi}_\lambda(a|z)\sum_{b=0}^{1}\left(\delta_{ab} - \widetilde{\pi}_\lambda(b|z)\right)\left(-\frac{1}{\beta}w_2(z)c(z,b)\right)$$

$$= \frac{w_2(z)}{\beta}\widetilde{\pi}_\lambda(a|z)\left[\left(\mathbb{E}_{a'\sim\widetilde{\pi}_\lambda(\cdot|z)}c(z,a')\right) - c(z,a)\right]. \tag{A.21}$$

Similarly, since (A.21) is bounded according to Assumptions 2 and 3, we can interchange the order of differentiation and expectation. Combining (A.17) and (A.21), (A.16) turns to be

$$d''(\lambda) = \beta - \mathbb{E}_{(z,l)\sim\rho}\frac{1}{\beta}w_2(z)^2\sum_{a=0}^{1}\widetilde{\pi}_\lambda(a|z)\left[\left(\mathbb{E}_{a'\sim\widetilde{\pi}_\lambda(\cdot|z)}c(z,a')\right) - c(z,a)\right]c(z,a)$$

$$= \beta + \frac{1}{\beta}\mathbb{E}_{(z,l)\sim\rho}w_2(z)^2\,\text{Var}_{a\sim\widetilde{\pi}_\lambda(\cdot|z)}c(z,a). \tag{A.22}$$

Again, due to Assumption 2 and 3, we have $c(z,a) \leq M$ and $w_2(z) \leq K$. Hence,

$$\beta \leq d''(\lambda) \leq \beta + \frac{M^2 K^2}{\beta}.$$

We therefore conclude that $d(\lambda)$ is strongly convex with parameter $\beta$, and its gradient is Lipschitz continuous with constant $\beta + \frac{M^2 K^2}{\beta}$. Then, by Theorem 2.1.5 in (Nesterov, 2013), choosing the step size as $\eta = 2/(\beta + \beta + M^2 K^2/\beta) = 2\beta/(M^2 K^2 + 2\beta^2)$ in (7), we have

$$|\lambda_{t+1} - \lambda^*| \leq \left( \frac{\beta + \frac{M^2 K^2}{\beta} - \beta}{\beta + \frac{M^2 K^2}{\beta} + \beta} \right)^t |\lambda_0 - \lambda^*| = \left( \frac{M^2 K^2}{M^2 K^2 + 2\beta^2} \right)^t |\lambda_0 - \lambda^*|. \tag{A.23}$$

In our setting, although the update involves a projection onto the feasible set, the projection operator is non-expansive. Consequently, standard results for projected gradient descent apply, and the projected updates achieve the same convergence rate as the corresponding unconstrained gradient descent.

**Step 3.** Finally, we show that $\pi_t$ converges to $\pi^*$ using the KL divergence.

Given $z$, the quantity $\mathrm{KL}(\pi_t(\cdot|z)\|\pi^*(\cdot|z))$ is a Bregman divergence between the corresponding natural parameters of the exponential family:

$$D_\psi(\theta_t, \theta^*) = \mathrm{KL}(\pi_t(\cdot|z)\|\pi^*(\cdot|z)),$$

where the convex potential $\psi$ is the log-partition function

$$\psi(\theta) = \log Q(z, -\theta\beta) = \log \sum_{a=0}^{1} \exp\left( \frac{1}{\beta} w_1(z) r(z,a,l) - \theta w_2(z) c(z,a) \right).$$

Here we let $\theta_t = -\lambda_t/\beta$ and $\theta^* = -\lambda^*/\beta$, treating $\widetilde{\pi}_\lambda(\cdot|z)$ as an exponential-family distribution with natural parameter $\theta$.

Then we have

$$\psi'(\theta) = \frac{\exp\left( \frac{1}{\beta} w_1(z) r(z,a,l) - \theta w_2(z) c(z,a) \right) c(z,a)}{\sum_{a'=0}^{1} \exp\left( \frac{1}{\beta} w_1(z) r(z,a') - \theta w_2(z) c(z,a') \right)} = \mathbb{E}_{a \sim \widetilde{\pi}_{-\theta\beta}(\cdot|z)} w_2(z) c(z,a).$$

By (A.22), we further obtain,

$$\psi''(\theta) = (-\beta)\left( -\frac{w_2(z)^2}{\beta} \mathrm{Var}_{a \sim \widetilde{\pi}_\lambda(\cdot|z)} c(z,a) \right) = w_2(z)^2 \, \mathrm{Var}_{a \sim \widetilde{\pi}_\lambda(\cdot|z)} c(z,a) \leq M^2 K^2.$$

By applying Taylor's expansion to $\psi$, there is

$$\psi(\theta_t) = \psi(\theta^*) + \psi'(\theta^*)(\theta_t - \theta^*) + \frac{1}{2}\psi''(\widetilde{\theta})(\theta_t - \theta^*)^2,$$

where $\widetilde{\theta}$ lies between $\theta_t$ and $\theta^*$. Consequently, we get

$$D_\psi(\theta_t, \theta^*) = \frac{1}{2}\psi''(\widetilde{\theta})(\theta_t - \theta^*)^2 \leq \frac{M^2 K^2}{2}(\theta_t - \theta^*)^2 = \frac{M^2 K^2}{2\beta^2}(\lambda_t - \lambda^*)^2,$$

which is equivalent to

$$\mathrm{KL}(\pi_t(\cdot|z)\|\pi^*(\cdot|z)) \leq \frac{M^2 K^2}{2\beta^2}(\lambda_t - \lambda^*)^2.$$

Averaging over $(z,l) \sim \rho$, we get

$$\mathrm{KL}(\pi_t\|\pi) = \mathbb{E}_{(z,l)\sim\rho}\mathrm{KL}(\pi_t(\cdot|z)\|\pi^*(\cdot|z)) \leq \frac{M^2 K^2}{2\beta^2}(\lambda_t - \lambda^*)^2.$$

Combining this with (A.23), we finally derive

$$\mathrm{KL}(\pi_t\|\pi) \leq \frac{M^2 K^2}{2\beta^2}\left( \frac{M^2 K^2}{M^2 K^2 + 2\beta^2} \right)^{2t}(\lambda_0 - \lambda^*)^2.$$

$\square$

# B. Experimental Details

## B.1. Data Construction and Evaluation Protocol

We construct all judge data using a unified generation protocol, and explicitly separate datasets used for training and in-distribution analysis from those reserved for out-of-distribution (OOD) evaluation.

**Judge Generation Protocol.** Across all datasets, we generate judge responses using the same base model from the QWEN3 family (1.7B/4B/8B), under two inference modes: non-reasoning and reasoning. All generations use an identical prompting format (Table C.3) and a fixed sampling temperature of 0.6. This ensures that any observed performance or cost differences are attributable solely to the inference mode and data distribution.

**Training and In-Distribution Data.** We construct the training corpus from three sources: SKYWORK-REWARD (Liu et al., 2024), MATH-STEP-DPO-10K (Lai et al., 2024), and CODE-PREFERENCE-PAIRS (Vezora, 2024). Specifically, we sample 20,000 instances from SKYWORK-REWARD, and an additional 20,000 instances from a mixture of MATH-STEP-DPO-10K and CODE-PREFERENCE-PAIRS. Together, these datasets cover general instruction-following, mathematical reasoning, and code-related preference judgments.

For each instance, we generate paired judge outputs under both inference modes. We record two signals for each mode: (i) a binary correctness indicator, obtained by comparing the judge's preference with the reference label provided by the dataset, and (ii) the number of tokens consumed during generation. These paired correctness and token-cost measurements define the empirical accuracy–cost trade-offs used both for the analysis in Section 2 and for training the routing policy. All training and in-distribution data are used exclusively for learning the routing policy.

**Out-of-Distribution Evaluation Data.** To evaluate generalization, we assess routing performance on multiple held-out benchmarks that are disjoint from the training data, including JUDGEBENCH (Tan et al., 2025), REWARDBENCH (Lambert et al., 2025), and REWARDBENCH-2 (Malik et al., 2025). These benchmarks span diverse evaluation domains and exhibit distributional shifts relative to the training corpus. We briefly summarize the domains they cover and the distribution in Table B.1.

For all OOD benchmarks, we apply the same judge generation protocol as used for training data, including the same base models, inference modes, prompts, and sampling temperature. Evaluation datasets are used solely for OOD testing and do not influence model selection or hyperparameter tuning.

## B.2. Text Representations

Each input instance is represented by a fixed-dimensional text embedding extracted via Qwen3-embedding-4B (Zhang et al., 2025b). Specifically, we encode the prompt and the two candidate responses separately, and concatenate their embeddings to form the policy input vector. The same representation procedure is applied consistently across all training and evaluation datasets.

## B.3. Routing Policy Architecture and Optimization

Our routing policy is a 4-layer MLP with ReLU activations and hidden widths $\{256, 128, 64\}$, mapping the input embedding to a single scalar logit.

Models are trained for 60 epochs using the AdamW optimizer with learning rate $10^{-4}$ and batch size 64. The dual variable associated with the budget constraint is updated using a step size of $10^{-3}$ and projected onto the non-negative reals after each update. An entropy regularization term with coefficient 0.005 is applied to encourage exploration and stabilize training.

For model selection, we evaluate checkpoints on a held-out validation set. Among checkpoints that satisfy the target budget constraint, we select the one achieving the highest validation accuracy. If no checkpoint satisfies the constraint, we select the checkpoint whose validation cost is closest to the target budget.

**Table B.1.** Domain distribution of OOD evaluation datasets.

| Benchmark | Domain | Count | Proportion | Description |
|---|---|---|---|---|
| JudgeBench (Tan et al., 2025) | Coding | 73 | 11.8% | Challenging programming questions from contest-style platforms (e.g., LeetCode, AtCoder, Codeforces). |
| | Math | 90 | 14.5% | Problems drawn from math competitions (e.g., AMC12, USAMO). |
| | Reasoning | 149 | 24.0% | Reasoning-focused evaluation set. |
| | Knowledge | 308 | 50.0% | College-level multiple-choice questions across 14 disciplines (e.g., Physics, Chemistry, Law), with up to 10 answer options. |
| RewardBench (Lambert et al., 2025) | Math | 447 | 15.0% | Aggregated from RewardBench original subsets. Subsets: *math-prm*. |
| | Coding | 984 | 33.0% | Subsets: *hep-cpp*, *hep-go*, *hep-java*, *hep-js*, *hep-python*, *hep-rust*. |
| | Chat | 814 | 27.3% | General assistant-style. Subsets: *alpacaeval-easy*, *alpacaeval-length*, *alpacaeval-hard*; *mt-bench-easy*, *mt-bench-medium*, *mt-bench-hard*; *llmbar-natural*, *llmbar-adver-neighbor*, *llmbar-adver-GPTInst*, *llmbar-adver-GPTOut*, *llmbar-adver-manual*. |
| | Safety | 740 | 24.8% | Refusal/safety-oriented. Subsets: *refusals-dangerous*, *refusals-offensive*, *xstest-should-refuse*, *xstest-should-respond*, *do-not-answer*. |
| RewardBench 2 (Malik et al., 2025) | Factuality | 475 | 26.9% | Evaluates detection of hallucinations and other basic factual errors in completions. |
| | Precise IF | 160 | 9.1% | Precise instruction following: judges whether outputs satisfy detailed, constraint-heavy instructions. |
| | Math | 183 | 10.4% | Assesses math ability on open-ended prompts spanning middle-school topics through college-level chemistry, calculus, and combinatorics. |
| | Safety | 450 | 25.5% | Measures appropriate compliance vs. refusal under harmful-use prompts and general safety-related behaviors. |
| | Focus | 495 | 28.1% | Tests whether models prefer high-quality, on-topic answers for general user queries. |

### B.4. Budget Settings and Repeated Trials

We evaluate the routing policy under a range of target compute budgets:

$$C \in \{2.0, 2.5, 3.0, 3.5, 4.0, 4.5, 5.0, 6.0, 7.0, 10.0\}.$$

For each budget level, we repeat the entire training and evaluation procedure 10 times with independent random seeds. We report the mean of both accuracy and realized cost ratio across these runs.

### B.5. Hyperparameter Tuning

**Optimization hyperparameters.** We select standard optimization hyperparameters via a grid search on an in-distribution validation set, and then fix them for all experiments. This includes the regularization parameter $\beta$, the learning rate ($10^{-4}$), batch size (64), the dual update step size ($10^{-3}$), and the entropy regularization coefficient (0.005).

**Temperature for robustness.** Since we assume no access to supervision for validation for OOD evaluation, we adopt the following rule-of-thumb. Let $\tau_R$ and $\tau_C$ denote the robustness strengths for reward and cost, respectively, where smaller values impose more aggressive (more robust) reweighting, while larger values indicate a more non-robust behavior. If the anticipated an OOD dataset requires less computation than in-distribution, we recommend using a smaller $\tau_R$ (stronger reward robustness) and a larger $\tau_C$ (weaker cost robustness) to better utilize the budget. If the anticipated an OOD dataset require more computation, we recommend using a smaller $\tau_C$ (stronger cost robustness) to reduce budget-violation risk,

**Table B.2.** Judge prompting templates for `instruct` vs. `reasoning` modes. Differences are highlighted in red.

| Field | Instruct mode | Reasoning mode |
|---|---|---|
| **System** | Please act as an impartial judge and evaluate the quality of the responses provided by two AI assistants to the user question displayed below.  You should choose the assistant that follows the user's instructions and answers the user's question better.  Your evaluation should consider factors such as the helpfulness, relevance, accuracy, depth, creativity, and level of detail of their responses. Begin your evaluation by comparing the two responses and provide **a short explanation** explanation.  Avoid any position biases and ensure that the order in which the responses were presented does not influence your decision.  Do not allow the length of the responses to influence your evaluation.  Do not favor certain names of the assistants.  Be as objective as possible.  After providing your explanation, output your final verdict by strictly following this format:  ``[[A]]'' if assistant A is better, ``[[B]]'' if assistant B is better. | Please act as an impartial judge and evaluate the quality of the responses provided by two AI assistants to the user question displayed below.  You should choose the assistant that follows the user's instructions and answers the user's question better.  Your evaluation should consider factors such as the helpfulness, relevance, accuracy, depth, creativity, and level of detail of their responses. Begin your evaluation by comparing the two responses and provide **an explanation**.  Avoid any position biases and ensure that the order in which the responses were presented does not influence your decision.  Do not allow the length of the responses to influence your evaluation.  Do not favor certain names of the assistants.  Be as objective as possible.  After providing your explanation, output your final verdict by strictly following this format:  ``[[A]]'' if assistant A is better, ``[[B]]'' if assistant B is better. |
| **User template** | [User Question]{question}

[The Start of Assistant A's Answer]{answer_a}[The End of Assistant A's Answer]

[The Start of Assistant B's Answer]{answer_b}[The End of Assistant B's Answer] | |

while keeping $\tau_R$ moderate to avoid over-conservatism. When the shift direction is uncertain, a conservative default is to use moderate robustness on both terms.

In the OOD evaluation in Section 5.3, we fix $\tau_R = 1$ and disable robust reweighting on the cost term.

## C. Supplementary Experimental Results

### C.1. Training Curves

In Figure C.1, we report the training curves for reward, realized cost, and lambda under different budgets. These training curves indicate that the Lagrange multiplier dynamically adapts to the tightness of the compute budget: it grows under stringent budgets to actively enforce cost control, but collapses toward zero as the budget loosens, signaling that the constraint becomes non-binding.

### C.2. Case Studies

In Table C.1, we report some representative cases to illustrate when and why reasoning helps.

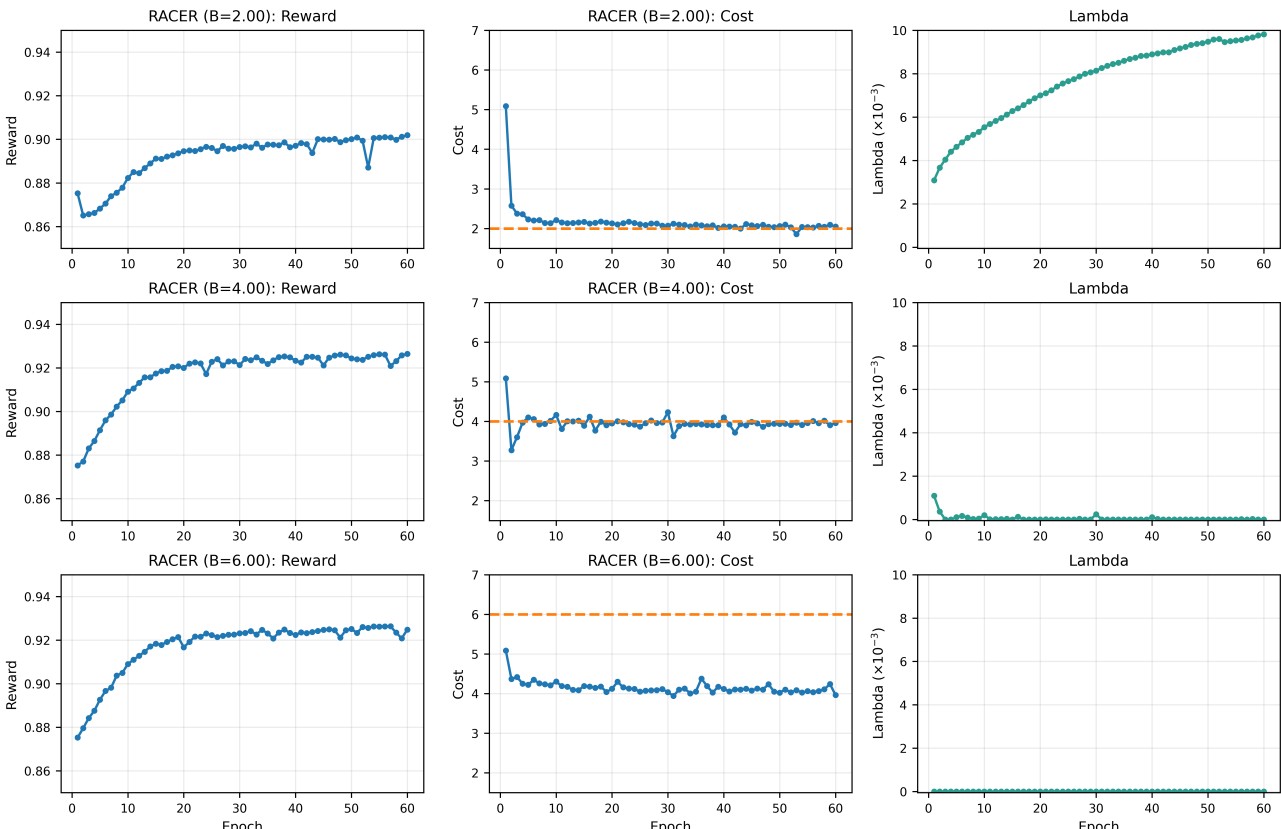

**Figure C.1.** Training dynamics of RACER under different cost budgets. Training reward (left), realized cost (middle), and the learned dual variable $\lambda$ (right) across epochs for budgets $B \in \{2, 4, 6\}$. The dashed line denotes the target cost budget. Reward increases monotonically while $\lambda$ adapts to the tightness of the constraint, driving the training cost toward the specified budget.

## C.3. Evaluation on Additional Model Family

To show that RACER is not specific to the Qwen3 family, we additionally train and evaluate RACER using DeepSeek-R1-Distill-Llama-8B (Guo et al., 2025) as the reasoning judge, and the corresponding Llama-3.1-8B-Instruct (Grattafiori et al., 2024) as the instruct mode judge. We follow the same data generation protocol defined in Section 5.3 and construct the training set from the mixture of SKYWORK-REWARD (Liu et al., 2024), MATH-STEP-DPO-10K (Lai et al., 2024) and CODE-PREFERENCE-PAIRS (Vezora, 2024). As shown in Table C.2, RACER consistently outperforms random routing at matched budgets and often surpasses all-reasoning accuracy at lower cost, showing that RACER has generalization with respect to the model family.

## C.4. Analysis of Self-Routing Behavior

Our approach learns the routing policy from data. As a comparison, we consider a simple self-routing baseline, where the LLM itself decides whether reasoning is needed. The training and evaluation data is the same as we used in Section 5.3, generated by Qwen3-4B pair. The self-routing baseline is implemented as a two-step procedure: we first prompt (see Table C.3) the instruct mode judge to decide whether to enable reasoning, and then evaluate the response generated under the chosen mode, measuring the corresponding accuracy and cost.

As shown in Table C.4, this strategy routes nearly all inputs to the reasoning mode, resulting in behavior and cost that are effectively equivalent to the all-reasoning judge. This suggests that self-routing fails to learn meaningful selectivity. In contrast, RACER learns to activate reasoning only when beneficial under a budget, achieving a better accuracy–cost trade-off.

**Table C.1.** Empirical case studies across benchmarks (excerpts only). Each row contrasts non-reasoning vs. reasoning judge outputs on the same instance; the label indicates whether the judge decision matches the ground-truth winner.

| Dataset (ID) | Non-Reasoning Judge | Reasoning Judge |
|---|---|---|
| JUDGEBENCH-MATH (515) | **INCORRECT (✗)**

*(excerpt)* Both Assistant A and Assistant B provide step-by-step solutions. The judge prefers Assistant B due to its more detailed explanations and justifications, which are considered more convincing and thorough, despite an error in counting the contribution of $49 = 7^2$. . . .

Final verdict:
[[B]] | **CORRECT (✓)**

*(excerpt)* The judge explicitly verifies the computation using the Legendre formula:

$$\left\lfloor \tfrac{50}{7} \right\rfloor + \left\lfloor \tfrac{50}{49} \right\rfloor = 7 + 1 = 8$$

It identifies that Assistant B undercounts the contribution of 49, and correctly determines that Assistant A provides the correct answer. . . .

Final verdict:
[[A]] |
| JUDGEBENCH-KNOWLEDGE (38) | **CORRECT (✓)**

*(excerpt)* Both Assistant A and Assistant B provided thorough and accurate responses to the user's question. . . .
Assistant B's response is more detailed and provides a step-by-step analysis of each option. They correctly identify the correct answers as (D), (G), and (I) but choose to focus on (D) as the main correct answer. . .
Additionally, Assistant B's response is more aligned with the user's request for a single correct answer.
Therefore, based on the evaluation, I would say that Assistant B's response is slightly better than Assistant A's response.
[[B]] | **INCORRECT (✗)**

*(excerpt)* Okay, so I need to evaluate which AI assistant, A or B, provided a better answer . . .
Assistant A's response . . . They provided explanations for each, pointing out which were correct and incorrect. For example, they correctly identified that option (I) is the right answer for vibration sense with a 128 Hz tuning fork. . . .
Assistant B's response . . . However, their reasoning had some issues. For instance, they mentioned that the assessment should focus on the dominant arm . . . and they didn't fully explain why 128 Hz is better than 256 Hz for vibration testing. They also made a mistake by suggesting that the correct answer is (D) when the user's main question was about vibration sense, which should point to (I) instead. . . . Assistant B, while correct on some points, didn't address the primary question accurately and had some inaccuracies in their reasoning. Therefore, Assistant A's response is more comprehensive and accurate.
[[A]] |
| RM-BENCH-CHAT (chat/404/0_0) | **CORRECT (✓)**

*(excerpt)* Both Assistant A and Assistant B provide a concise and relevant answer. Assistant A correctly states that Mars appears reddish due to iron oxide . . . On the other hand, Assistant B incorrectly states that Mars appears reddish due to copper oxide . . .
Considering the accuracy of the responses, Assistant A's answer is more reliable and trustworthy. . . .
Therefore, based on the evaluation, the final verdict is:
[[A]] | **CORRECT (✓)**

*(excerpt)* Okay, let's see. I need to evaluate the responses . . .
Assistant A's response . . . It's concise and accurate.
Assistant B's response . . . Both responses mention reddish and rust, but Assistant B says copper oxide instead of iron oxide. That's a mistake. . . .
So, in terms of accuracy, Assistant A is correct, whereas Assistant B is wrong. . . . However, the accuracy is key here. Since Assistant A provided the right information, it's better. . . .
[[A]] |

**Table C.2.** Routing performance and cost comparison across budgets using data generated by the DeepSeek-R1-Distill-Llama-8B and Llama-3.1-8B-Instruct pair.

| Budget | RACER Acc (%) | RACER Cost | Random Acc (%) | Random Cost |
|---|---|---|---|---|
| All-Instruct | 67.32 | 1.00 | – | – |
| 2.0 | 79.37 | 2.05 | 73.01 | 2.44 |
| 2.5 | 80.41 | 2.45 | 74.99 | 2.88 |
| 3.0 | 80.87 | 2.89 | 76.06 | 3.15 |
| 3.5 | 80.83 | 3.20 | 76.98 | 3.39 |
| All-Reasoning | 79.93 | 4.12 | – | – |

**Table C.3.** User prompt template used for self-routing (instruct mode).

| Template | Content |
|---|---|
| **User Prompt** | You are about to judge which of two AI responses better answers a user question.
[User Question]{question}
[The Start of Response A]{answer_a}[The End of Response A]
[The Start of Response B]{answer_b}[The End of Response B]
Do you need to enable your reasoning mode (i.e., think step-by-step carefully) to judge which response is better for the above question and responses?  Answer with a single word:  yes or no. |

**Table C.4.** Self-routing behavior across model scales in Qwen3 family.

| Model | Reasoning Fraction | Accuracy (%) | | | Cost Ratio |
|---|---|---|---|---|---|
| | | **All-instruct** | **All-reasoning** | **Self-routing** | |
| 1.7B | 99.91 | 70.04 | 78.35 | 78.34 | 19.07 |
| 4B | 99.51 | 81.96 | 86.50 | 86.50 | 6.25 |
| 8B | 99.88 | 82.20 | 87.86 | 87.86 | 6.36 |

