# OpenReview forum: "Reasoning Is Not Free: Robust Adaptive Cost-Efficient Routing for LLM-as-a-Judge"
_ICML.cc/2026/Conference — ICML 2026 regular_

### Official Review · Reviewer_er84 · 2026-03-11

**Soundness:** 3
**Presentation:** 3
**Significance:** 3
**Originality:** 3
**Overall Recommendation:** 4
**Confidence:** 2

**Summary:**

This paper introduces the Robust Adaptive Cost-Efficient Router (RACER), a framework designed to optimize the deployment of reasoning-capable LLMs in automated evaluation (LLM-as-a-Judge) by mitigating the high token costs and "overthinking" tendencies of explicit reasoning on simpler tasks. Unlike standard routers that degrade under out-of-distribution (OOD) shifts, RACER formulates the routing decision as a constrained distributionally robust optimization (DRO) problem using a KL-divergence uncertainty set to dynamically balance evaluation accuracy and computational cost. Solved via a primal-dual algorithm with mathematically proven optimal policy uniqueness and linear convergence, RACER consistently outperforms non-robust baselines on standard benchmarks, achieving superior accuracy-cost trade-offs while strictly adhering to budget constraints under real-world data distribution shifts.

**Compliance With Llm Reviewing Policy:**

Affirmed.

**Final Justification:**

My question has been resolved.

**Key Questions For Authors:**

See weakness

**Limitations:**

yes

**Strengths And Weaknesses:**

## Strengths
1. Clear and well-motivated problem formulation: The paper identifies an important practical gap in LLM-as-a-Judge, reasoning can improve judgment on structured verification tasks such as math and coding, but it also incurs much higher cost and can even hurt on simpler cases.

2. Strong empirical analysis of when reasoning helps:The paper controlled comparison between paired reasoning and non-reasoning judges, matched by model family and evaluated on multiple judge benchmarks. This makes the paper’s conclusion about heterogeneous accuracy-cost trade-offs more convincing.

3. Methodology supported by theoretical analysis and empirical evaluation: The paper formulates the task as a constrained distributionally robust optimization (DRO) problem. The authors provide theoretical proofs for the optimal policy's uniqueness and the algorithm's linear convergence , and they use out-of-distribution (OOD) experiments to demonstrate the method's behavior under distribution shift and budget constraints.

## Weakness
1. Potential discrepancy between theoretical assumptions and the experimental model: While the paper provides detailed proofs for the uniqueness of the optimal policy and linear convergence, these results are built on the assumption that the policy class $\Pi$ is convex. However, the experiments utilize a four-layer MLP as the router, which is a non-convex model. It would be helpful if the authors could clarify whether the theoretical guarantees are expected to hold for such non-convex architectures.

---

> ### Author Rebuttal · Authors · 2026-03-30
>
> We are very grateful for your valuable comments. Below, we summarize your comments in quotes and provide our point-by-point responses.
>
> > W1: Potential discrepancy between theoretical assumptions and the experimental model: While the paper provides detailed proofs for the uniqueness of the optimal policy and linear convergence, these results are built on the assumption that the policy class $\Pi$ is convex. However, the experiments utilize a four-layer MLP as the router, which is a non-convex model. It would be helpful if the authors could clarify whether the theoretical guarantees are expected to hold for such non-convex architectures.
>
> We thank the reviewer for the insightful comment. Our theoretical analysis is conducted over a convex policy class, which allows us to establish the *uniqueness of the optimal policy and linear convergence guarantees*. This formulation is natural in our router framework, where router policies are viewed as distributions over prompt–response pairs. In practice, to handle large vocabulary spaces, we parameterize the router policy using a four-layer MLP. While this does not exactly match the convex policy class assumed in theory, it’s commonly applied in reinforcement learning and optimization, where parameterized models are used to approximately represent the optimal policy characterized in the theoretical analysis.
>
> Importantly, our convergence results can be **extended to the parameterized (non-convex) setting under additional assumptions on the approximation quality of the policy class $\Pi$**. In particular, if each policy update approximates the optimal entropy-regularized solution up to a controlled error, then the last-iterate convergence result continues to hold up to an additional approximation error term, as commonly studied in prior work (e.g., [1, 2]).
>
> Furthermore, as shown in Fig. C.1 (Appendix C.2), the training curves are pretty stable across multiple budget settings, providing empirical evidence that the learned router policy exhibits consistent convergence behavior in practice.
>
> To provide more supplementary experiment results which demonstrate the overall robust and better performance of the proposed method in various settings, we further added **DeepSeek-R1 results** in our response to *Reviewer 61dW*, **additional baseline comparisons** in our response to *Reviewer WrMp*, and **reasoning activation patterns** in our response to *Reviewer nXNs*.
>
>
> [1] Ding, Dongsheng, et al. "Last-iterate convergent policy gradient primal-dual methods for constrained mdps." Advances in Neural Information Processing Systems 36 (2023): 66138-66200.
>
> [2] Zhan, Wenhao, et al. "Policy mirror descent for regularized reinforcement learning: A generalized framework with linear convergence." SIAM Journal on Optimization 33.2 (2023): 1061-1091.

---

> > ### Author Rebuttal · Reviewer_er84 · 2026-04-01
> >
> > Thank you for your response, my question has been resolved

---

> > > ### Author Response · Authors · 2026-04-03
> > >
> > > Thank you for your time and for considering our rebuttal. We sincerely appreciate your thoughtful feedback and are pleased that our responses helped clarify the paper.

---

### Official Review · Reviewer_nXNs · 2026-03-11

**Soundness:** 3
**Presentation:** 3
**Significance:** 3
**Originality:** 2
**Overall Recommendation:** 4
**Confidence:** 3

**Summary:**

This paper studies whether reasoning improves LLM-as-a-Judge performance. The authors show empirically that reasoning inference can improve judgment accuracy on tasks requiring structured verification (e.g., math and coding), but often brings limited gains on simpler tasks while incurring higher computational cost. To address this trade-off, the paper proposes RACER, a routing method that dynamically selects between reasoning and non-reasoning inference under a compute budget.

**Compliance With Llm Reviewing Policy:**

Affirmed.

**Final Justification:**

The authors' clarifications and experiments have addressed my main concerns. The additional experiments will enhance my understanding of the features that trigger reasoning, making the work more convincing.

**Key Questions For Authors:**

1. The paper motivates the work using the question of reasoning transfer from problem-solving to judging. How does the proposed routing method help us understand or measure this transfer?
2. How sensitive is the method to the choice of the KL-divergence uncertainty set used in the robustness formulation?

**Limitations:**

Yes

**Strengths And Weaknesses:**

### Strengths
1. The paper presents an empirical study comparing reasoning and non-reasoning inference modes when LLMs are used as automated judges.
2. The paper studies the cost-accuracy trade-off in LLM-as-a-Judge systems. The problem is practical and relevant as LLM-based evaluation becomes more widely used.
3. The framework is simple and can be applied on top of existing hybrid reasoning models without modifying the underlying judge model.

### Weaknesses
1. The introduction motivates the work by asking whether reasoning abilities learned from problem-solving tasks transfer to judgment tasks. However, the proposed method does not directly study this transfer mechanism. Instead, it mainly learns a routing policy to allocate reasoning under a cost constraint. As a result, the connection between the motivating question and the proposed algorithm is somewhat weak.
2. Routing between expensive and cheaper inference modes under a budget constraint has been explored in prior work on LLM routing and cost-aware inference.
3.  The paper does not analyze what features actually trigger reasoning mode in the learned router. Understanding these patterns could provide useful insight into when reasoning is truly necessary.

---

> ### Author Rebuttal · Authors · 2026-03-30
>
> We are very grateful for your valuable comments. Below, we summarize your comments in quotes and provide our point-by-point responses.
>
> > W1: The paper is motivated by whether reasoning transfers from problem solving to judgment, but the method mainly learns a routing policy. The connection is somewhat weak.
>
> Our work is motivated by whether reasoning transfers from problem solving to judgment, which we study empirically in Secs. 2.1 and 2.2. The results show that reasoning can substantially improve judgment but its benefits are heterogeneous and often come at higher cost. This motivates our method: *rather than modeling the transfer mechanism itself, we learn a routing policy that selectively activates reasoning under a token budget*. To handle query shifts at deployment, we further develop a distributionally robust routing policy for cost-aware judgment. Overall, while transfer is our starting point and is examined empirically, our main focus is the practical challenge of cost-aware, distributionally robust routing.
>
> > W2: Routing between expensive and cheaper inference modes under a budget constraint has been explored.
>
> To the best of our knowledge, RACER is the **first** study to investigate *routing in the LLM-as-a-Judge setting*. It is also the **first** to address *distribution shift in LLM routing while providing theoretical guarantees*. In addition, many existing methods [1,2] rely on weighted quality-cost trade-offs or thresholding, rather than optimizing under *an explicit hard budget constraint*. This further distinguishes RACER from prior work.
>
> > W3: The paper does not analyze what features actually trigger reasoning mode in the learned router.
>
> To better understand what features trigger reasoning, we **conduct two complementary analyses**.
>
> First, we analyze **reasoning activation with respect to the relative benefit of reasoning**, partitioning examples into four groups: both correct, only reasoning correct, only instruct correct, and both wrong (as in Fig. 2). On both ID testing and OOD testing (RewardBench-v2), reasoning is activated most often when it provides unique benefit (only reasoning correct) or when the decision is hardest (both wrong), and least often when instruct mode already suffices. This suggests that the router selectively allocates reasoning where it is most useful.
>
> | Dataset | Budget | Both Correct % | Only Reasoning Correct % | Only Instruct Correct % | Both Wrong % |
> |---|---|---|---|---|---|
> | ID | 2 | 24.7 | 49.2 | 13.8 | 21.9 |
> | ID | 3 | 43.1 | 74.3 | 40.3 | 58.6 |
> | ID | 4 | 55.8 | 78.4 | 48.9 | 62.65 |
> |||||||
> | OOD | 2.0 | 9.9 | 16.2 | 10.0 | 13.6 |
> | OOD | 3.0 | 52.4 | 66.5 | 51.4 | 63.7 |
> | OOD | 4.0 | 69.4 | 78.3 | 66.7 | 77.2 |
>
> Second, we study **the semantic similarity between the candidate responses**. We find that the *reasoning activation is the highest when the two responses are highly similar*. Intuitively, these are cases where the choice is more subtle and may require deeper reasoning to distinguish between close alternatives.
>
> | Response-pair embedding similarity bin | B=2 | B=3 | B=4 |
> |---|---|---|---|
> | Q1 | 8.2 | 33.5 | 39.6 |
> | Q2 | 12.7 | 41.0 | 51.8 |
> | Q3 | 39.0 | 57.9 | 69.3 |
> | Q4 | 44.8 | 56.4 | 71.9 |
>
> Together, these results suggest that the learned router captures meaningful patterns related to *problem difficulty and selectively allocates reasoning when it is most likely to help*.
>
> > Q1: How does the proposed routing method actually help us understand or measure the transfer of reasoning ability from problem-solving tasks to judgment tasks?
>
> As discussed in our response to W1, our method is not designed to explicitly model the transfer mechanism itself; rather, our method leverages the observed transfer behavior which is *heterogeneous across domains and often comes with a higher cost* to address the problem of **when and how** to selectively activate reasoning under cost constraints in a principled way.
>
> > Q2: How sensitive is the method to the choice of the KL-divergence uncertainty set in the robustness formulation?
>
> Our method is not conceptually restricted to the KL-divergence uncertainty set. More broadly, the robust-routing formulation can be easily extended with other f-divergence-based uncertainty sets as well, since what we need is an uncertainty set that bounds the deviation between the deployment distribution and the empirical training distribution. We adopt the KL-divergence set in this work mainly because it yields a tractable worst-case reweighting formulation, which leads to a practically efficient optimization procedure.
>
> Besides the results above, we further **added DeepSeek-R1 results** in our response to *Reviewer 61dW* and **additional baseline comparisons** in our response to *Reviewer WrMp*.
>
> [1] Song, Wei, et al. "Irt-router: Effective and interpretable multi-llm routing via item response theory."
>
> [2] Ong, Isaac, et al. "RouteLLM: Learning to Route LLMs from Preference Data."

---

> > ### Author Rebuttal · Reviewer_nXNs · 2026-04-03
> >
> > Thanks to the authors for answering my questions. I will raise my score from 3 to 4.

---

> > > ### Author Response · Authors · 2026-04-03
> > >
> > > Thank you for your time and for considering our rebuttal. We sincerely appreciate your thoughtful feedback and are pleased that our responses helped clarify the paper.

---

### Official Review · Reviewer_WrMp · 2026-03-12

**Soundness:** 2
**Presentation:** 3
**Significance:** 2
**Originality:** 2
**Overall Recommendation:** 4
**Confidence:** 4

**Summary:**

This paper studies whether reasoning-mode LLMs are actually better judges than non-reasoning variants (controlled other factors such as model size etc). They propose RACER, a budget-constrained router that uses distributionally robust optimization over a KL-divergence uncertainty set to decide when to use reasoning. The empirical results to introduce the issues are convincing.

**Compliance With Llm Reviewing Policy:**

Affirmed.

**Final Justification:**

The rebuttal partially addressed my concerns and thus I raise the score a little bit.

**Key Questions For Authors:**

1. There are many LLM routing solutions available. It is unclear for me why these models not be included in baseline comparison.
2. The numerical results are not very convincing. Have you considered adding additional results to show it works?
3. Reasoning routing is a popular topic. Many papers discuss relevant issues in various degrees, the key innovation is not very clear though I really like the authors presentation. Could you clarify the significance and innovation of the manuscript, especially compared with many existing work in LLM routing.

**Limitations:**

Yes.

**Strengths And Weaknesses:**

Strengths:
1. The study is well designed and well written.
2. The method is straightforward and seems quite novel.

Weaknesses:
1. There is no real benchmarking compared with existing methods. LLM routing is not a new area, even though the framework of reasoning or not is somewhat new here, at least some comparisons with existing methods should be included. Some popular works have also been ignored. Some powerful methods such as IRT-Router or others that based on text embedding as the inputs may need to be added as baseline.
2. In the paper, the authors use the text embedding to build the covariates and the routing model. We can also ask model to decide reasoning or not automatically (for example, QWen 3 default Hybrid Reasoning mode). The authors should at least show the trained method can have a much better performance compared with default automate hybrid reasoning mode. Use the random guess as a baseline seems like a toy example. IRT-Router seems to be one strong candidate, but feel free to use others popular ones as baseline.
3. The ablation study is limited. The authors do not check the impact of each component in ablation study.

---

> ### Author Rebuttal · Authors · 2026-03-30
>
> We are very grateful for your valuable comments. Below, we summarize your comments in quotes and provide our point-by-point responses.
>
> > W1: The paper lacks benchmarking against established routing methods.
>
> We have **added comparisons with three representative baselines: RouterBench-KNN, RouteLLM-MF, and M-IRT** under the same budget constraint (B=4), with results averaged over 10 replications.
>
> | Method | 1.4b Acc % | 1.4b Cost | 4b Acc % | 4b Cost | 8b Acc % | 8b Cost |
> |---|---:|---:|---:|---:|---:|---:|
> | RACER | 72.24 | 3.60 | 85.79 | 3.38 | 89.95 | 3.90 |
> | RouterBench-KNN | 71.27 | 2.58 | 84.11 | 2.51 | 86.84 | 2.61 |
> | RouteLLM-MF | 69.42 | 3.75 | 84.69 | 3.43 | 88.20 | 4.11 |
> | M-IRT | 71.60 | 3.37 | 84.26 | 2.65 | 88.89 | 3.43 |
>
> As shown in the table, **RACER achieves the best accuracy across all three model families while remaining within budget**. For 1.4b, 4b, and 8b, RACER improves over the strongest baseline by **+0.64, +1.10, and +1.06**, respectively. These results suggest that RACER can more effectively exploit the available budget, allocating more budget when beneficial and yielding a stronger reward-cost trade-off than prior routing methods. We also note that RACER is a general framework, and can in principle be combined with more advanced routing architectures to further improve performance.
>
> > W2: Since some models already support automatic hybrid reasoning, the paper should compare RACER to default reasoning modes.
>
> We thank the reviewer for the suggestion. Qwen3 Hybrid Reasoning supports user-controlled switching between thinking and non-thinking modes, rather than automatic routing. We therefore construct a self-routing baseline that first asks the model whether reasoning is needed. We observe that this baseline routes nearly all inputs to the reasoning mode, and as a result, its cost and behavior are essentially equivalent to the reasoning judge, corresponding to *the green point in the top-right of Figure 4*. In contrast, RACER learns selective routing under a budget and activates reasoning only when beneficial, achieving a better accuracy–cost trade-off.
>
> | Model | Self-routing Acc % | All-instruct Acc % |  All-reasoning Acc % | Self-routing Cost Ratio |  Reasoning Activation Fraction |
> |---|---|---|---|---|---|
> |1.7B|78.34|70.04|78.35|19.07|99.91|
> |4B|86.50|81.96|86.50|6.25|99.51|
> |8B|87.86|82.20|87.86|6.36|99.88|
>
> > W3: The ablation study is too limited.
>
> We thank the reviewer for this comment. **Sec. 5.2 provides a component-wise ablation to isolate reward and cost robustness**. The results show that these two components play different roles under different OOD settings (Fig. 3).
>
> We then provide **an additional ablation on $\beta$ (weight for entropy regularization) at 3 representative budget levels**. Removing it hurts accuracy under the tight budget, confirming its value when the constraint is binding. Performance remains stable for $\beta\in{0.005,0.01}$, while $\beta=0.05$ consistently degrades accuracy, supporting the necessity and our default choice of $\beta$.
>
> | $\beta$ | B=2 Acc % | Cost | B=3 Acc % | Cost | B=4 Acc % | Cost |
> |---|---------|----------|---------|----------|---------|----------|
> | 0 | 85.23 | 1.93 | 86.71 | 2.92 | 86.82 | 3.69 |
> | **0.005 (default)** | 85.54 | 2.01 | 86.71 | 2.92 | 86.69 | 3.55 |
> | 0.01 | 85.50 | 1.95 | 86.68 | 2.91 | 86.71 | 3.77 |
> | 0.05 | 84.81 | 1.97 | 85.98 | 2.88 | 86.22 | 3.82 |
>
> > Q1: Why are existing LLM routing methods not included in the baseline comparisons?
>
> Many prior routing methods handle budget only at inference time, rather than explicitly optimizing under a hard cost constraint, so they are not directly aligned with our formulation and were not included initially. Moreover, our method is a general framework for budget-constrained routing and can be combined with more advanced router architectures. Our initial comparison therefore focused on validating the framework itself. That said, we agree that comparisons with representative works are valuable, and we have added such baselines above.
>
> > Q2: Additional results?
>
> Besides the results above, we further **added DeepSeek-R1 results** in our response to *Reviewer 61dW* and **analyzed reasoning activation patterns** in our response to *Reviewer nXNs*.
>
> > Q3: What is the key innovation and significance of this paper?
>
> To the best of our knowledge, RACER is the **first** study to investigate *routing in the LLM-as-a-Judge setting*. It is also the **first** to address *distribution shift in LLM routing while providing theoretical guarantees*. In addition, many existing methods [1,2] rely on weighted quality-cost trade-offs or thresholding, rather than optimizing under *an explicit hard budget constraint*. This further distinguishes RACER from prior work.
>
> [1] Song, Wei, et al. "Irt-router: Effective and interpretable multi-llm routing via item response theory."
>
> [2] Ong, Isaac, et al. "RouteLLM: Learning to Route LLMs from Preference Data."

---

> > ### Author Rebuttal · Reviewer_WrMp · 2026-04-01
> >
> > Thank you for your detailed response. The results are not very convincing. Routing in the end is a tradeoff between accuracy and cost. The proposed method does not show it can beat existing methods. I raise the score since the team is honest about their work and the overall idea is kind of interesting.

---

> > > ### Author Response · Authors · 2026-04-03
> > >
> > > Thank you again for your time and for considering our rebuttal. We sincerely appreciate your thoughtful feedback and are glad that our responses helped clarify the paper.
> > >
> > > We would just like to gently clarify one point. We agree that routing ultimately involves a tradeoff between accuracy and cost. Our main goal is to study this tradeoff in a constrained and realistic setting, where routing must be optimized under a hard budget and potential distribution shift. RACER is designed to explicitly address these two aspects within a unified framework, whereas many prior routing methods may not consider them jointly. We also view RACER as a general training framework rather than a fixed router architecture, and it can be naturally combined with more advanced router models.
> > >
> > > Thank you again for your constructive comments. We will revise the paper to clearly reflect the additional evaluations and clarifications discussed during the review process.

---

### Official Review · Reviewer_61dW · 2026-03-13

**Soundness:** 3
**Presentation:** 2
**Significance:** 2
**Originality:** 3
**Overall Recommendation:** 4
**Confidence:** 3

**Summary:**

Strengths：
1. The paper provides a nuanced empirical characterization showing that explicit reasoning is not a universal panacea, as it can lead to negative gains on simple tasks and significantly higher costs.
2. The paper provides rigorous mathematical proofs for the uniqueness of the optimal policy and linear convergence of the iterates, offering a level of theoretical support previously missing in llm routing literature.
3. Experiments demonstrate that RACER successfully concentrates reasoning on the instances where it provides the largest marginal benefit, matching the accuracy of an all-reasoning judge at roughly half the computational cost.

Weaknesses:
1. The proposed RACER framework relies on a KL-divergence uncertainty set that simulates distribution shifts strictly by reweighting existing training samples. This creates a fundamental support set limitation: as proven in Theorem 3.1, the worst-case distributions must share the same support as the empirical training data. Consequently, the router's robustness is confined to domains represented in the training set and may fail when encountering queries from fundamentally novel domains or disjoint feature spaces.
2. The current binary action space is overly rigid, which restricts cost-efficiency; transitioning toward a granular routing strategy—such as dynamic token budgeting or model cascading—presents a key opportunity to optimize the system's performance and resource allocation.
3. While RACER claims to be adaptive, tuning its core robustness parameters ($\tau_R$ and $\tau_C$) heavily relies on manually predicting future distribution shifts using a rule-of-thumb. In real-world LLM deployments, query distributions fluctuate unpredictably, making manual guessing impractical. This reliance on human intuition—along with the use of a sub-optimal moderate default when shifts are unknown—severely undermines the system's autonomy and contradicts its claim of true adaptiveness.

Question：
1. How sensitive is RACER's performance if the deployed environment experiences completely unpredictable, rapidly fluctuating, or even adversarial distribution shifts (where the rule-of-thumb guidelines cannot be applied)?
2. Could the current optimization objective be easily adapted to include latency and memory bandwidth as penalty constraints, rather than relying solely on token usage?
3. While the evaluation across different scales (1.7B, 4B, and 8B) of the Qwen3 family provides a good initial proof of concept, the reliance on a single model family significantly limits the generalizability of the empirical claims. Models from different architectures—such as DeepSeek-R1 distillations or Llama-based reasoning models—may exhibit fundamentally different accuracy-cost trade-off behaviors and baseline judging capabilities. To convincingly demonstrate that the RACER framework is architecture-agnostic and broadly applicable, could the authors supplement the experiments by evaluating their routing policy on at least one additional, distinct reasoning model family?

**Compliance With Llm Reviewing Policy:**

Affirmed.

**Final Justification:**

I have read the rebuttal and it fully resolved my concerns. I have witten them in the below acknowledgement, and I will maintain my score.

**Key Questions For Authors:**

see summary

**Limitations:**

yes

**Strengths And Weaknesses:**

see summary

---

> ### Author Rebuttal · Authors · 2026-03-30
>
> We are very grateful for your valuable comments. Below, we summarize your comments in quotes and provide our point-by-point responses.
>
> > W1: RACER’s robustness is limited by its KL-divergence uncertainty set, as it only reweights training samples and cannot handle novel or out-of-support domains.
>
> We agree that KL-based uncertainty sets require source and target distributions to have overlapping support, and thus do not provide robustness guarantees for entirely novel domains with disjoint support. This limitation is NOT unique to RACER, but is *inherent to distributionally robust optimization*. More broadly, this is consistent with theoretical results in the domain adaptation literature that *adaptation to an arbitrary target domain is impossible without additional assumptions relating source and target* [1]. Accordingly, RACER is designed for the practically relevant setting where deployment domains exhibit distribution shifts but still share support with training data.
>
> > W2: The binary routing design is too rigid; finer-grained strategies like dynamic token budgeting or model cascading could improve efficiency.
>
> The proposed RACER is not restricted to binary actions: its robust constrained formulation can be naturally extended to a finite action space, where actions could correspond to different reasoning strategies or cascade combinations. We focus on the binary setting as this paper specifically studies the reasoning-versus-non-reasoning scenario.
>
> > W3: Although RACER is presented as adaptive, its robustness still relies on heuristic assumptions about future shifts.
>
> Thank you for raising this point. We believe “adaptive” may have been misunderstood: in our paper, it refers to **instance-wise routing under a budget constraint**, NOT assumption-free robustness to future shifts. Our method targets practically relevant deployment shifts that still share support with the training data. In such settings, prior knowledge about plausible shifts and the target robustness level can help inform parameter selection.
>
> > Q1: How does RACER perform under unpredictable, rapidly changing, or adversarial shifts where heuristic parameter choices fail?
>
> Figure 4 (JudgeBench) shows that RACER continues to outperform random guessing even under substantial distribution shifts, which indicates that it remains reasonably robust under OOD conditions. However, RACER is primarily designed for realistic deployment shifts, rather than fully unpredictable, rapidly fluctuating, or adversarial shifts. In such highly non-stationary settings, the current uncertainty set may not provide universal guarantees, and we will clarify this scope in the revision. An important direction for future work is to strengthen robustness in these settings, for example, by incorporating non-stationary bandit techniques [2,3] that use change-point detection and recent deployment feedback to adapt routing. Such mechanisms could make RACER better suited to rapidly changing environments.
>
> > Q2: Can the optimization objective be extended to account for system-level constraints such as latency and memory bandwidth, instead of focusing only on token usage?
>
> Yes, the current formulation is not tied to token usage; token cost is simply the most natural and easy-to-measure resource in our setting. More generally, the constrained optimization framework can accommodate other deployment constraints such as latency and memory bandwidth as long as they are measurable. In that sense, extending RACER beyond token-based budgeting is conceptually straightforward.
>
> > Q3: Can the authors show that RACER is architecture-agnostic by evaluating it on at least one additional reasoning-model family beyond Qwen3?
>
> We **additionally tested RACER on DeepSeek-R1-Distill-Llama-8B**, using Llama-3.1-8B-Instruct as the corresponding instruct-mode model. As shown in the table below, RACER consistently outperforms random routing under matched budgets, and in several settings even exceeds all-reasoning accuracy at substantially lower cost. These results provide additional evidence that RACER is not specific to the Qwen3 family and can generalize to another model family.
>
> | Budget | RACER Acc % ↑ | RACER Cost ↓ | Random Acc % | Random Cost |
> | --- | --- | --- | --- | --- |
> | All-Instruct | 67.32 | 1.0 | - | - |
> | 2.0 | 79.37 | 2.05 | 73.01 | 2.44 |
> | 2.5 | 80.41 | 2.45 | 74.99 | 2.88 |
> | 3.0 | 80.87 | 2.89 | 76.06| 3.15 |
> | 3.5 | 80.83 | 3.20 | 76.98 | 3.39 |
> | All-Reasoning | 79.93| 4.12 | - | - |
>
> Besides the results above, we added **additional baseline comparisons** in our response to *Reviewer WrMp* and analyzed **reasoning activation patterns** in our response to *Reviewer nXNs*.
>
> [1] David, Shai Ben, et al. "Impossibility theorems for domain adaptation."
>
> [2] Wu, Qingyun, Naveen Iyer, and Hongning Wang. "Learning contextual bandits in a non-stationary environment."
>
> [3] Cao, Yang, et al. "Nearly optimal adaptive procedure with change detection for piecewise-stationary bandit."

---

> > ### Author Rebuttal · Reviewer_61dW · 2026-04-03
> >
> > Thanks for the detailed response. Most my concerns have been resolved, and I will keep my positive score.

---

> > > ### Author Response · Authors · 2026-04-03
> > >
> > > Thank you for your time and for considering our rebuttal. We sincerely appreciate your thoughtful feedback and are pleased that our responses helped clarify the paper.

---

### Decision · Program_Chairs · 2026-04-30

**Decision:**

Accept (regular)

**Comment:**

While reviewers raised a few key concerns during the paper's initial round of review (such as lack of model diversity and strong baselines in experiments), the authors successfully addressed these concerns during the rebuttal with additional experimental results. Multiple reviewers really liked the combination of theoretical analysis and practically useful experimental results, which the AC also appreciates. Overall, the review team agrees with the paper's novel contributions and thorough experiments (assuming that the authors will add the additional experiments as promised to the final version). The review team agrees that the paper can be accepted to the conference.